# Sectoral attribution of greenhouse gas and pollutant emissions using multi-species eddy covariance on a tall tower in Zurich, Switzerland

Rainer Hilland<sup>1</sup>, Josh Hashemi<sup>1,2</sup>, Stavros Stagakis<sup>3</sup>, Dominik Brunner<sup>4</sup>, Lionel Constantin<sup>4</sup>, Natascha Kljun<sup>5</sup>, Ann-Kristin Kunz<sup>1</sup>, Betty Molinier<sup>5</sup>, Samuel Hammer<sup>6</sup>, Lukas Emmenegger<sup>4</sup>, and Andreas Christen<sup>1</sup>

**Correspondence:** Rainer Hilland (rainer.hilland@meteo.uni-freiburg.de)

#### Abstract.

Eddy covariance measurement of species that are co-emitted with carbon dioxide ( $\rm CO_2$ ), such as carbon monoxide ( $\rm CO_2$ ) and nitrogen oxides NO and NO<sub>2</sub> ( $\rm NO_x$ ), provides an opportunity to attribute a net flux to individual source or sink categories. We present eight months of continuous simultaneous measurements of fluxes (F) of  $\rm CO_2$ ,  $\rm CO$ ,  $\rm NO_x$ , methane ( $\rm CH_4$ ), and nitrous oxide ( $\rm N_2O$ ) from a tall tower (112 m agl) in Zurich, Switzerland. Median daily fluxes of  $F_{\rm CO_2}$  were 1.47 times larger in the winter (Nov-Mar) as opposed to summer (Aug-Oct) months (10.9 vs. 7.4  $\mu$ mol m<sup>-2</sup> s<sup>-1</sup>); 1.08 times greater for  $F_{\rm CO}$  (30 vs. 28 nmol m<sup>-2</sup> s<sup>-1</sup>); 1.08 times greater for  $F_{\rm NO_x}$  (14 vs. 13 nmol m<sup>-2</sup> s<sup>-1</sup>); 1.01 times greater for  $F_{\rm CH_4}$  (13.5 vs. 13.3 nmol m<sup>-2</sup> s<sup>-1</sup>); and not statistically significantly different for  $F_{\rm NO_2}$ . Flux ratios of  $F_{\rm CO}/F_{\rm CO_2}$  and  $F_{\rm NO_x}/F_{\rm CO_2}$  are well characterised by inventory emission ratios of stationary combustion and road transport in cold months. In warm months both  $F_{\rm CO}/F_{\rm CO_2}$  and  $F_{\rm NO_x}/F_{\rm CO_2}$  systematically exceed expected inventory ratios during the day, while no statistically significant seasonal difference is observed in  $F_{\rm NO_x}/F_{\rm CO_2}$ , indicating biospheric photosynthetic activity. A linear mixing model is proposed and applied to attribute half-hourly  $F_{\rm CO_2}$ ,  $F_{\rm CO}$ , and  $F_{\rm NO_x}$  to stationary combustion and road transport emission categories as well as determine the biospheric  $F_{\rm CO_2}$ . Flux attribution is reasonable at certain times and from certain wind directions, but over-attributes CO and NO<sub>x</sub> fluxes to road traffic and CO<sub>2</sub> fluxes to stationary combustion, and overestimates photosynthetic CO<sub>2</sub> uptake.

### 1 Introduction

The majority of the world's population now lives in urban areas, with the projected proportion to reach 68% by 2050. In the European Union, this proportion already exceeds 70% (United Nations, 2019). Cities are estimated to cause 70% of global energy-related greenhouse gas emissions (Lwasa et al., 2022) despite covering only 3% of the Earth's land surface (Liu et al., 2014). Cities are therefore crucial for emission mitigation initiatives (European Commission, 2021; Lwasa et al., 2022), and

<sup>&</sup>lt;sup>1</sup>Environmental Meteorology, Institute of Earth and Environmental Sciences, University of Freiburg, Germany

<sup>&</sup>lt;sup>2</sup>Alfred Wegener Institute, Helmholtz Centre for Polar and Marine Research, Potsdam, Germany

<sup>&</sup>lt;sup>3</sup>Environmental Sciences, University of Basel, Switzerland

<sup>&</sup>lt;sup>4</sup>Empa, Swiss Federal Laboratories for Materials Science and Technology, Dübendorf, Switzerland

<sup>&</sup>lt;sup>5</sup>Center for Environment and Climate Science, Lund University, Sweden

<sup>&</sup>lt;sup>6</sup>Institute for Environmental Physics, University of Heidelberg, Germany

some cities are already pursuing ambitious plans for reduction of carbon dioxide (CO<sub>2</sub>) emissions above and beyond national-level plans (e.g., Stadt Zürich, 2023). To understand emissions at the city-scale, many cities have developed emission inventories based on bottom-up approaches in which emissions are estimated based on local activity data such as traffic counts and fuel usage. Estimates in the form of emission inventories are temporally coarse, and rely on a series of assumptions and scaling factors to downscale yearly bulk emissions to spatially and temporally resolved estimates, with large associated errors and uncertainties (Simon et al., 2008; Gately and Hutyra, 2017; Gurney et al., 2021).

Independent monitoring of total CO<sub>2</sub> emissions and the attribution of emissions to specific sectors is critical to determine the efficacy of emission reduction efforts and policy changes (Wu et al., 2016; Lauvaux et al., 2020). Eddy covariance (EC) flux measurements are currently the best available method for providing direct and time-resolved in situ measurements of the vertical exchange of scalars in the atmosphere. Although urban areas violate the fundamental assumption of surface homogeneity that is inherent to estimating long-term exchange irrespective of wind direction (Aubinet et al., 2012), EC has been successfully applied in cities for over two decades to quantify dynamics of CO<sub>2</sub> emissions and monitor long- and short-term emission changes (e.g., Grimmond et al., 2002; Christen et al., 2011; Stagakis et al., 2019).

However, the EC method integrates all sources and sinks of  $CO_2$  within its source area, making sectoral attribution of fossil  $CO_2$  or separation of fossil (ff) and biospheric (bio)  $CO_2$  difficult. Spatially and temporally resolved high-resolution emission inventories combined with footprint modelling have been used to attribute integral fluxes of  $CO_2$  to emission sectors (e.g., Christen et al., 2011; Stagakis et al., 2023). Another promising approach is that of co-emitted species ratios (Nathan et al., 2018) in which the ratio between  $CO_2$  and another species is used to attribute measured fluxes to specific emission processes or to separate bio and  $ffCO_2$ , e.g., by using the co-emitted species carbon monoxide (CO) and the  $CO/ffCO_2$  ratio. Diurnal changes in concentrations of  $CO/ffCO_2$  have been shown to correspond to diurnal changes in road traffic (Turnbull et al., 2015) in a city where CO emissions were predominantly traffic related. There is currently no method to measure  $ffCO_2$  directly with the frequency needed to calculate EC fluxes. However, if the  $CO/ffCO_2$  emission ratio is known then the measured flux ratios of CO and total  $CO_2$  may be used to partition the  $CO_2$  flux into biospheric and fossil components (Wu et al., 2022). Further, if the sector-specific  $CO/ffCO_2$  ratio is known, it may be possible to partition integral EC fluxes to specific emission sectors. City emission inventories therefore offer intriguing possibilities to estimate sector-specific emission ratios of  $CO_2$  and co-emitted species, which may be used in turn to attribute integral EC fluxes to specific sectors at a half-hourly scale without the need for individual footprint modelling.

Urban EC has mostly focused on  $CO_2$  measurements (Feigenwinter et al., 2012; Matthews and Schume, 2022). However, the EC method may be applied to any scalar measured with sufficient frequency. CO is a common urban pollutant and short-lived greenhouse gas produced by, among other processes, incomplete combustion of hydrocarbons, and is commonly co-emitted during combustion along with  $CO_2$ . Due to its role as a pollutant, CO is commonly measured by air quality networks as lower-frequency (30 min to 1 h) average concentrations. Flux measurements of CO via EC are rare. A few studies have been conducted over agricultural sites (e.g., Cowan et al., 2018; Murphy et al., 2023) and CO is occasionally measured in urban environments as part of multi-species campaigns, e.g., alongside  $NO_x$  (Karl et al., 2017),  $CH_4$  (Helfter et al., 2016), or  $N_2O$  (Famulari et al., 2010). Nitrogen oxides ( $NO_x$ ) are other common urban pollutants created in combustion processes. Urban EC

studies of  $NO_x$  are similarly scarce but have revealed significant underestimation of traffic emissions in selected inventories (Marr et al., 2013; Lee et al., 2015; Vaughan et al., 2016; Karl et al., 2017). Methane (CH<sub>4</sub>) is a potent greenhouse gas also emitted through human activity. While methane is released by combustion processes, it is less suitable as a tracer for fossil fuel emissions given its abundant non-combustion sources like leakage of natural-gas distribution networks, and non-fossil sources like landfills and agriculture (Nathan et al., 2018). Nevertheless, due to its strong global warming potential, urban methane emissions are receiving increasing attention, though only a few direct urban EC studies exist (Helfter et al., 2016; Pawlak and Fortuniak, 2016; Stichaner et al., 2024). Nitrous oxide ( $N_2O$ ) is another long-lived potent greenhouse gas emitted through anthropogenic activities. Similarly to  $CH_4$ , it is less suited as an ff $CO_2$  tracer as there are many non-fossil production mechanisms of  $N_2O$ , and anthropogenic  $N_2O$  emissions are overwhelmingly agricultural on a global scale (Nathan et al., 2018; Tian et al., 2020). However,  $N_2O$  fluxes may be a useful tracer for fossil-fuel emissions within cities themselves where they have been shown to scale well with road traffic (Famulari et al., 2010) and have shown no seasonal variability in a semi-urban site (Järvi et al., 2014).

This work has three research goals:

- To characterise the diurnal patterns of fluxes of CO<sub>2</sub>, CO, NO<sub>x</sub>, CH<sub>4</sub>, and N<sub>2</sub>O in Zurich, and quantify possible seasonal differences;
  - 2. to compare measured ratios of CO<sub>2</sub>, CO, and NO<sub>x</sub> to a local emissions inventory; and
  - 3. to investigate whether net fluxes of CO<sub>2</sub>, CO, and NO<sub>x</sub> can be attributed to source categories using a linear mixing model.

In Sec. 3.1 we address research goal 1 and present simultaneous eddy covariance fluxes of the aforementioned species measured on a tall tower near the city centre of Zurich, Switzerland, using a single instrument employing laser absorption spectroscopy. To our knowledge this is the first time that EC fluxes of these species have been measured together. Aggregated diurnal and seasonal fluxes of each species are shown. In Sec. 3.2 we address research goal 2 and present ratios of the three most common and well-correlated species (CO<sub>2</sub>, CO, and NO<sub>x</sub>). Finally, in Sec. 3.3 we present a linear mixing model for partitioning total fluxes into source sectors and for estimating the biogenic CO<sub>2</sub> flux from observed species flux ratios and their expected ratios from the inventory.

## 2 Methods

70

## 2.1 Study Location and Site Description

The city of Zurich is the largest city in Switzerland with approximately 443,000 inhabitants in the city proper and approximately 1.6 million inhabitants in the wider metropolitan area. The city is situated at the northern edge of Lake Zurich at approximately 400 m asl in a valley surrounding the Limmat river running roughly SE-NW. The valley floor is relatively flat with a gentle slope towards the NW. The valley edges reach about 600 m asl to the N/NE and about 850 m asl to the S/SW (Fig. 2).

The city has roughly 55,000 individual buildings and 236,000 residential units as of 2024. Heating is primarily via natural gas ( $\approx 110,000$  units), district heating ( $\approx 50,000$  units) and oil ( $\approx 45,000$  units). A small number of residential units ( $\approx 3,000$ ) use wood heating. The city's economy is almost entirely service-oriented, aside from limited infrastructure such as a waste incineration plant which lies to the far NE of the city limits. The city's vehicle fleet comprises approximately 131,000 personal cars of which 74,000 are gasoline, 34,000 diesel, 15,000 hybrid, and 7,000 electric. Significant non-personal transport occurs within and through the city. The city proper has approximately 70,000 urban trees excluding the forested areas (roughly 25% of the city) of which about 90% are broadleaf deciduous trees and 10% are coniferous trees. The city's climate is oceanic according to the Köppen climate classification (Köppen: Cfb) with an annual mean temperature of 9.3 °C. All data on buildings, heating, vehicle fleet, and trees, is obtained from the city of Zurich climate and/or statistics webpages (Stadt Zürich, 2025).

## 2.2 Instrumentation and Measurements

100

105

110

115

120

Eddy covariance (EC) measurements were carried out from August 2022 to March 2023 roughly 1.5 km NW of the city centre on a communications antenna on the top of a high-rise apartment building (Bullingerstrasse 73: 47° 22' 52" N, 8° 30' 26" E) built in the late 1970s. The building itself is 95 m tall and square in plan form with sides of 22.8 m. The instruments were installed 17 m above rooftop level on the antenna. The building is significantly taller than the surrounding buildings, the mean height of which is 13.3 m  $\pm$ 8 m in a 1.5 km radius, though the building is in a cluster of four nearly identical high-rise residential buildings set 43 to 68 m apart with heights 76 m (S), 85 m (SW), and 66 m (NNW). The displacement height within this 1.5 km radius is 23.4 m according to the parameterisation of Kanda et al. (2013).

The installation consisted of one integrated open-path eddy covariance system (OPEC) and a high-frequency closed-path (CPEC) multi-species gas analyser. The measured gases and other variables from these two instruments are summarised in Table 2.2. The open-path IRGASON (Campbell Scientific, Logan UT, USA) integrates a 3-dimensional sonic anemometer-thermometer with an open-path infrared gas analyser that measures  $CO_2$  and  $H_2O$  densities. Both wind and gas measurements were acquired at 20 Hz. The IRGASON was mounted 111.8 m agl and oriented towards W (270° from N). Logging was performed on a Campbell Scientific CR6 datalogger (Campbell Scientific, Logan UT, USA).

The closed-path analyser was an MGA7 (Multi-species Gas Analyser,-7 species, MIRO Analytical, Wallisellen, Switzerland) located in a machine room on the roof-level of the high-rise with an inlet co-located with the IRGASON. The analyser was placed in a temperature-controlled enclosure for temperature stability. The inlet was connected to the analyser via approximately 33 m of Synflex tubing (8 mm outer diameter, 5.6 mm inner diameter). The MGA7 measured 7 different gas species (summarised in Table 2.2) simultaneously using mid-infrared quantum cascade laser spectroscopy (Li et al., 2013) at high frequencies suitable for EC measurements (nominally 10.3 Hz). The MGA7 measured all gases as a wet molar fraction (mol mol<sup>-1</sup>). A NeoDry30E-2DBK (Kashiyama Industries, Saku City, Nagano Japan) dry vacuum pump downstream of the analyser drew sample air through the analyser at approximately 10 SL min<sup>-1</sup>.

The suitability of the instrument for measuring turbulent fluxes of the selected species is demonstrated in Fig. 1, in which average normalised spectra and co-spectra for each species are shown against the normalised frequency and theoretical slopes according to Kaimal et al. (1972). As reference, the normalised spectrum and co-spectrum are also shown for the OPEC

**Table 1.** Variables and descriptions measured from the open-path (Campbell Scientific IRGASON) and closed-path (MIRO MGA7) EC systems. Precision and accuracy figures are as per manufacturer user manuals.

| OPEC (Ca              | OPEC (Campbell Scientific IRGASON) |                        |                                       |                                        |                        | CPEC (Miro MGA7)                |                                     |                  |  |  |
|-----------------------|------------------------------------|------------------------|---------------------------------------|----------------------------------------|------------------------|---------------------------------|-------------------------------------|------------------|--|--|
| Variable              | Description                        | Units                  | Precision                             | Accuracy                               | Variable               | Description                     | Units                               | Precision (1 Hz) |  |  |
| $u_x$                 | Longitudinal wind                  | ${\rm m~s^{-1}}$       | $1~\mathrm{mm~s^{-1}}$                | ±2%                                    | $\chi_{\rm CO_2}$      | CO <sub>2</sub> molar fraction  | $\mathrm{mol}\ \mathrm{mol}^{-1}$   | 1 ppm            |  |  |
| $u_y$                 | Lateral wind                       | $\rm m\;s^{-1}$        | $1 \; \mathrm{mm} \; \mathrm{s}^{-1}$ | $\pm 2\%$                              | $\chi_{\rm H_2O}$      | H <sub>2</sub> O molar fraction | $\bmod  \mathrm{mol}^{-1}$          | 40 ppm           |  |  |
| $u_z$                 | Vertical wind                      | ${\rm m}~{\rm s}^{-1}$ | $0.5\;\mathrm{mm\;s^{-1}}$            | $\pm 2\%$                              | χco                    | CO molar fraction               | $\bmod  \mathrm{mol}^{-1}$          | 0.4 ppb          |  |  |
| $T_s$                 | Sonic temperature                  | °C                     | 0.025 °C                              |                                        | χno                    | NO molar fraction               | $\mathrm{mol}\ \mathrm{mol}^{-1}$   | 0.8 ppb          |  |  |
| $T_a$                 | Air temperature                    | °C                     |                                       | $\pm 0.15~^{\circ}\mathrm{C}$          | $\chi_{ m NO_2}$       | NO <sub>2</sub> molar fraction  | $\bmod  \mathrm{mol}^{-1}$          | 0.25 ppb         |  |  |
| p                     | Air pressure                       | kPa                    |                                       | $\pm 1.5~\mathrm{kPa}$                 | $\chi_{\rm N_2O}$      | $N_2\mathrm{O}$ molar fraction  | $\bmod  \bmod^{-1}$                 | 0.5 ppb          |  |  |
| $ ho_{\mathrm{CO}_2}$ | CO <sub>2</sub> density            | ${\rm mg~m^{-3}}$      | $0.2~\mathrm{mg~m}^{-3}$              | $\pm 0.1\%$ °C <sup>-1</sup>           | $\chi_{\mathrm{CH_4}}$ | CH <sub>4</sub> molar fraction  | $\mathrm{mol} \; \mathrm{mol}^{-1}$ | 1 ppb            |  |  |
| $ ho_{ m H_2O}$       | H <sub>2</sub> O density           | ${\rm g~m}^{-3}$       | $3.5~{\rm g}~{\rm m}^{-3}$            | $\pm 0.3\%  {}^{\circ}\mathrm{C}^{-1}$ |                        |                                 |                                     |                  |  |  |

Figure 1. Normalised spectra and co-spectra for each species measured by the CPEC as well as the  $CO_2$  from the OPEC. The (co-)spectra are plotted against the normalised frequency. Theoretical slopes according to Kaimal et al. (1972) are plotted as dashed black lines.

instrument, which is a standard instrument for eddy-covariance studies. The spectra are taken from all unstable periods which pass quality control (see Sec. 2.3) and have an above-average absolute flux for the given species. The influence of spectral attenuation in the inlet tubing is visible in all species above normalised frequencies of  $\approx$ 60 (natural frequency  $\approx$ 2 Hz). The average  $N_2O$  spectrum demonstrates noise at normalised frequencies greater than 4 (natural frequency 0.1 Hz). The normalised co-spectrum with vertical wind fluctuations is shown in the right panel of Fig. 1. Good agreement with the theoretical slope of -4/3 and agreement with the OPEC is shown for all species, though at the noise-equivalent frequencies a noticeable deviation in  $N_2O$  exists at similar frequencies to those shown by (Järvi et al., 2014).

**Figure 2.** Location of measurement site (yellow triangle) over a land cover map of Zurich (Amtliche Vermessung). Isolines up to the 80% footprint climatology according to Kljun et al. (2015) are overlaid in white. Note that the footprint is modelled after all quality control filtering and represents only the measurement periods retained for analysis. The blue box centred on the tower shows the 4 x 4 km<sup>2</sup> box centred on the tower from which reference inventory molar emission ratios are determined (Sec. 2.4) and covers 67% of the study period's footprint.

Fig. 2 shows the measurement location marked with a yellow triangle and land cover classification along with the full campaign 80% source area for the flux measurements derived with the flux footprint model of Kljun et al. (2015). When weighting by the footprint, the dominant surface cover is vegetation (38%). Residential buildings comprise 10% while industrial and commercial buildings cover 3.5% and 4%, respectively. Roads are 9% of the footprint and rails 7.4%. The final 25% is categorised as *other*, namely bare or paved surfaces such as foot paths.

Frequency of counts by wind direction (%)

Figure 3. Wind regime for half-hour periods during the observation period which pass the CO<sub>2</sub> quality control tests.

Wind characteristics at the measurement site and for the measurement period are shown in the wind roses in Fig. 3, separated by summer/autumn months (Aug-Oct, left) and winter/spring (Nov-Mar, right). In this work the terms summer or warm months will be used to refer to the Aug 1, 2022 - Oct 31, 2022 measurement period and the terms winter or cold months will be used for the Nov 1, 2022 - Mar 31, 2023 measurement period interchangeably. The chosen seasonal split broadly corresponds with changes in local emissions due to heating degree days as well as the change of local time from Central European Daylight Savings Time (CEDT, UTC+2) to Central European Standard Time (CET) on the morning of 30 October 2022. In the summer months (left) the wind is more frequently from the lake (SE) and calmer than the winter months, with a mean wind speed of 2.9 m s<sup>-1</sup>. In the winter months the wind is trimodal along an axis perpendicular to the valley, with one mode coming from the SW and another from the NE, and generally higher wind speeds with a mean of 3.6 m s<sup>-1</sup>.

## 2.3 Processing and Quality Control

135

The OPEC (IRGASON) and CPEC (MGA7) systems were sampled at different rates which necessitated the merging of these two datasets. To create a harmonious dataset for EC calculations, the gas measurements of the MGA7 were upsampled to 20 Hz and merged with the IRGASON measurements using a nearest-neighbour approach with a 50 ms search window. Both systems were synchronised to UTC via an NTP server: The MGA7 once per week and the IRGASON logger every 24h. To account for unexpected clock drift and travel time of the sample gas through the inlet to the MGA7, the systems were synchronised by finding the time lag of maximum correlation between the 20 Hz CO<sub>2</sub> signals. Time lags for individual 30-

minute averaging periods in which correlation was poor (correlation coefficient < 0.5) due to low IRGASON signal strength were linearly interpolated to correct clock drift and then determined via covariance maximisation with the vertical wind using a search window of  $\pm 0.5$  s. The median time lag was 4.15 s.

MGA7 measurements were compared against flask sample measurements obtained from a relaxed eddy accumulation (REA) sampler co-located at the same measurement inlets as the CPEC. The REA sampler used fast-switching valves to conditionally sample up- and down-draft winds as determined from the OPEC system: one flask for updrafts and one flask for downdrafts, which were subsequently analysed at the ICOS Flask and Calibration Laboratory in Jena, Germany. More information on the details of the REA flask sampler and inter-instrument comparison may be found in Kunz et al. (2025). Each discrete pair (one up-flask and one down-flask) of REA samples was analysed to determine the per-species difference in concentrations in up- and down-draft flasks:  $\Delta \chi_{\rm Flask} = \chi_{\rm Flask, up} - \chi_{\rm Flask, down}$ . As the instruments were synchronised in time, the measurements from the MGA7 at the same timestamps could also be averaged and the corresponding difference in measurements from the CPEC ( $\Delta \chi_{\rm MGA7} = \chi_{\rm MGA7, up} - \chi_{\rm MGA7, down}$ ) could also be determined. For the eddy-covariance method, because high-frequency measurements are Reynolds-averaged, it is only necessary that there is no bias in high-frequency measurements and therefore measurements are robust if  $\Delta \chi_{\rm MGA7}/\Delta \chi_{\rm Flask} \approx 1$ . Over 80 reference flask measurement pairs we found an average  $\Delta \chi_{\rm MGA7}/\Delta \chi_{\rm Flask}$  of 1.003 for  $\chi = {\rm CO}_2$ ; 1.007 for  $\chi = {\rm CO}_3$ ; 1.008 for  $\chi = {\rm CH}_4$ ; and 1.053 for  $\chi = {\rm N}_2{\rm O}$ , corresponding to a respective average flux error of 0.3% for  $\chi = {\rm CO}_3$ ; 0.7% for  $\chi = {\rm CO}_3$ ; 0.8% for  $\chi = {\rm CH}_4$ ; and 5.0% for  $\chi = {\rm CH}_4$ . This calibration approach was only possible for non-reactive species.






For the reactive species NO and NO<sub>2</sub> the CPEC was regularly (initially every few weeks until Jan. 2023 and thereafter every 11.5 hours) zeroed against an NO- and NO<sub>2</sub>-free 50 L tank of reference air (Synthetische Luft 5.6, PanGas AG, Dagmersellen, Switzerland). No in situ measurement was available to validate the span of the NO and NO2 measurements. A comparison of the average concentration of NO and NO2 against measurements from local air quality measurement stations revealed a systematic underestimation of these species by the MGA7. While photochemical interactions may affect concentrations of NO and  $NO_2$  between emission sources and the measurement tower, total  $NO_x$  should be conserved on the short (3-5 minutes) time scales between emission and measurement considered here (Lee et al., 2015; Vaughan et al., 2016). Therefore the surface NO<sub>x</sub> concentrations were calculated at a local air quality measurement station (Zurich Kaserne, 47° 22' 39.23", 8° 31' 49.54") located in a nearby park away from heavy emission sources such as traffic, and the total  $NO_x$  concentration from the MGA7 was calculated by summing NO and NO2 molar fractions. The raw MGA7 measurements were scaled against the Kaserne measurements using a linear best fit model ( $r^2 = 0.65$ , RMSE 6.1 ppb) of hourly-averaged concentrations under day-time unstable conditions with wind speeds >3 m s<sup>-1</sup>, assuming that vertical and horizontal concentration gradients are small under these conditions. The slope correction of 2.33 was applied to the raw MGA7 concentrations and a total NO<sub>x</sub> flux calculated. From future measurement campaigns with the instrument using an identical inlet setup and a co-located reference instrument without this inlet, we estimate approximately 20% of the underestimation to be a systematic problem caused by the use of a steel inlet tube. The remaining bias is assumed to be an error in spectroscopic setup and retrieval functions. The scaling against a surface reference station introduces significant uncertainties associated with both reference station measurements as well as atmospheric transport, and the reader is cautioned to consider this during the results and discussion.

Open path measurements can be affected by erroneous spikes due to objects interfering (e.g. rain, dirt, insects along the optical path) with the path. To remove problematic data but retain real increases in mixing ratios, a despiking algorithm was run over the time series of vertical wind, CO<sub>2</sub>, and H<sub>2</sub>O. Despiking of the open-path measurements of the IRGASON was conducted using a modification of the median absolute deviation method described by Mauder et al. (2013), namely using the upper and lower limits defined by Mauder et al. (2013) but keeping observations in which 3 or more consecutive outliers occur to account for the large observed skewness of gas and temperature measurements typically observed in cities. No despiking was performed on the closed path signals from the MGA7. Regular inspection of raw data did not reveal any data points that would be considered spikes, and the environmental concerns that tend to cause spiky measurements in the open-path data do not affect the closed-path system.

Calculation of turbulent fluxes  $F_{\chi}$  was conducted via the EddyPro software (v7.0.9, Li-COR Biosystems) for 30-minute blocks. Coordinate rotation was performed via double rotation and turbulent fluctuations defined by block average (Rebmann et al., 2012). The Moncrieff et al. (1997) correction for high-pass filtering effects was used for both the OPEC and CPEC fluxes. To account for high-frequency spectral attenuation in the CPEC, the low-pass correction according to Fratini et al. (2012) was applied. The low-pass spectral correction according to Moncrieff et al. (2004) was applied to the OPEC.

Initial quality control was performed using the 0-1-2 flag system of Mauder and Foken (2004) in which data are categorised as 0 - best quality data suitable for fundamental research, 1 - acceptable quality data for observation programs, and 2 - low quality data. This system incorporates two tests: the steady state test and the integral turbulence characteristics (ITC) test, both according to Foken and Wichura (1996). Each gas species is tested separately. Data flagged either 0 or 1 were included in the present analysis; data flagged 2 were excluded.





The quality flag for flux ratios (e.g.,  $F_{\rm CO}/F_{\rm CO_2}$ ) was set to the highest flag of either component species, i.e., a period in which the  $F_{\rm CO}$  flag is 0 and the  $F_{\rm CO_2}$  flag is 1 will result in  $F_{\rm CO}/F_{\rm CO_2}$  flag 1. Winds from E were distorted by the antenna, mounting equipment, and IRGASON instrument body. Therefore periods in which the mean vector wind direction over 30 minutes was between 70 and 100 degrees from N were excluded. A friction velocity  $(u_*)$  filter was also implemented. Friction velocity filters are an accepted way to remove periods in which measured EC fluxes tend to decouple from surface fluxes (Aubinet et al., 2012) though their usage in urban EC, where friction velocities tend to be greater, is debated (Matthews and Schume, 2022). An examination of  ${\rm CO_2}$  fluxes binned by  $u_*$  showed significant and systematic decrease in flux magnitude at  $u_* < 0.2$  m s<sup>-1</sup> however, and therefore these observations were removed. Angles of attack exceeding  $|20^{\circ}|$  were removed. Finally, periods in which there were known issues with the MGA7 instrument due to software or hardware errors were removed. Fluxes were aggregated based on hourly binning and therefore no attempt at gap-filling or quantification of error or uncertainty for individual flux periods was performed. The storage flux was calculated using the default single-point estimation of EddyPro as no profile measurements below the tower were available. The storage flux for all species followed a typical pattern (positive overnight and negative during the day). The average diurnal storage flux for  ${\rm CO_2}$  was -0.1  $\mu$  mol m<sup>-2</sup> s<sup>-1</sup> with an inter-quartile range of -3.3 to 2.7  $\mu$ mol m<sup>-2</sup> s<sup>-1</sup>. The storage flux for CO was 0.0 (-8.0 to 7.2) nmol m<sup>-2</sup> s<sup>-1</sup>, for  ${\rm NO_x}$  0.0 (-3.6 to 3.4) nmol m<sup>-2</sup> s<sup>-1</sup>, for  ${\rm CH_4}$  0.0 (-12.0 to 9.6) nmol m<sup>-2</sup> s<sup>-1</sup>, and for  ${\rm N_2O}$  0.0 (-0.2 to 0.2) nmol m<sup>-2</sup> s<sup>-1</sup>.

**Table 2.** Number and percentage of retained measurement periods for each species and species pair considered in this work, separated by warm (Aug-Oct) and cool (Nov-Mar) months.

|                            | Aug       | g-Oct     | Nov-Mar   |           |  |
|----------------------------|-----------|-----------|-----------|-----------|--|
| Species                    | Incl. (n) | Incl. (%) | Incl. (n) | Incl. (%) |  |
| $F_{\rm CO_2}$             | 2674      | 61        | 4398      | 61        |  |
| $F_{ m H_2O}$              | 2391      | 54        | 3779      | 52        |  |
| $F_{\mathrm{CO}}$          | 2777      | 63        | 4498      | 62        |  |
| $F_{ m NO_x}$              | 2769      | 63        | 4500      | 62        |  |
| $F_{ m N_2O}$              | 2656      | 60        | 3736      | 51        |  |
| $F_{\mathrm{CH_4}}$        | 2338      | 53        | 3945      | 54        |  |
| $F_{\rm CO}/F_{{ m CO}_2}$ | 2542      | 58        | 4207      | 58        |  |
| $F_{ m NO_x}/F_{ m CO_2}$  | 2536      | 57        | 4203      | 58        |  |
| $F_{ m NO_x}/F_{ m CO}$    | 2576      | 58        | 4167      | 57        |  |
| $F_{\mathrm{CO}_2}$ (OPEC) | 2296      | 52        | 3926      | 54        |  |

The study period of 1 August 2022 to 31 March 2023 contains 11 663 half-hour flux averaging periods. The distribution of data quality is summarised in Table 2. Retention for the MGA7 species over both warm and cool months ranges from 52% ( $N_2O$  and  $H_2O$ ) to 61% ( $CO_2$ , CO,  $NO_2$ ). This is similar to the average retention value found in other urban EC studies (Matthews and Schume, 2022, Table 1), though the wide range of different quality control regimes creates a large spread in retention rates. A large number of filtered periods are instrument downtime, both planned and unplanned, and reduce the data retention by 7-8%. The wind direction filter subsequently removed 253 periods (2.2%). The  $u_*$  filter removed a significant number of periods: between 7 and 10% depending on the species. Data retention is fairly comparable between seasons, differing by only a few percent. One exception is  $F_{N_2O}$ , for which technical problems resulted in multiple weeks of lost measurements during Nov-Mar.

## 2.4 Emission Inventory




An emission inventory for 2022 was created by the city of Zurich on a GIS format which provides yearly total emissions of various species and pollutants for the city of Zurich (Brunner et al., 2025). The Swiss Laboratory for Air Pollution / Environmental Technology of EMPA has processed this inventory according to standard Gridded Nomenclature for Reporting (GNFR) categories at  $100 \times 100 \text{ m}^2$  spatial resolution.

Within a 4 x 4 km<sup>2</sup> square centred on the tower ( $\approx$ 60-70% of the flux footprint for the study period) sectoral contributions as derived from the emission inventory are dominated by road transport and other stationary combustion (Table 3), accounting for 81% of CO emissions, 91% of non-respiration  $CO_2$  emissions, 84% of  $NO_x$  emissions, 86% of  $CH_4$  emissions, and 92% of  $N_2O$  emissions. Human respiration likely contributes an additional 10% of  $CO_2$  and may be non-negligible for  $CH_4$  (Brunner

**Table 3.** Emissions inventory overview for a 4 x 4 km<sup>2</sup> box centred on the measurement tower. Categories given with their GNFR letter. Minor categories have been omitted.

| Category                        | $CO_2$ (Gg yr <sup>-1</sup> ) | $CO (Mg yr^{-1})$ | $NO_x (Mg yr^{-1})$ | $\mathrm{CH_4} \ (\mathrm{Mg} \ \mathrm{yr}^{-1})$ | $N_2O~(Mg~yr^{-1})$ |
|---------------------------------|-------------------------------|-------------------|---------------------|----------------------------------------------------|---------------------|
| Public Power (A)                | 15.6                          | 1.5               | 9.7                 | 1.7                                                | 0.03                |
| Industry (B)                    | 6.6                           | 7.2               | 8.6                 | 0.6                                                | 0.06                |
| Other Stationary Combustion (C) | 206.5                         | 145.1             | 72.3                | 20.4                                               | 1.01                |
| Road Transport (F)              | 71.6                          | 158.9             | 130.6               | 3.5                                                | 2.60                |
| Offroad (I)                     | 5.1                           | 34.7              | 18.5                | 0.1                                                | 0.21                |
| Waste (J)                       | 0.3                           | 13.4              | 0.7                 | 1.6                                                | < 0.01              |

**Table 4.** Molar ratios of select species and categories for the 4 x 4 km<sup>2</sup> area (All, NE, SE, SW, and NW) around the measurement tower as well as for the entire area of the city of Zurich: spatial average weighted by total mass of emissions and separated by major wind direction. Symbols are those defined in Eqs. 4 to 9.

| G :                  | G .                   | 0 1 1        | Ratio |      |      |      |      |           | T I 14                   |
|----------------------|-----------------------|--------------|-------|------|------|------|------|-----------|--------------------------|
| Species              | Category              | Symbol       | All   | NE   | SE   | SW   | NW   | Full City | Units                    |
| CO/CO <sub>2</sub>   | Road transport        | $a_{\rm rt}$ | 3.49  | 3.04 | 3.57 | 4.38 | 3.70 | 3.95      | mmol mol <sup>-1</sup>   |
|                      | Stationary combustion | $a_{\rm sc}$ | 1.10  | 1.28 | 0.95 | 1.40 | 0.94 | 1.19      | mmol mol <sup>-1</sup>   |
| $\mathrm{NO_x/CO_2}$ | Road transport        | $b_{ m rt}$  | 1.74  | 1.62 | 1.77 | 1.81 | 1.88 | 1.88      | mmol mol <sup>-1</sup>   |
|                      | Stationary combustion | $b_{ m sc}$  | 0.33  | 0.35 | 0.31 | 0.36 | 0.34 | 0.35      | mmol mol <sup>-1</sup>   |
| NO <sub>x</sub> /CO  | Road transport        | $c_{ m rt}$  | 0.50  | 0.54 | 0.49 | 0.41 | 0.51 | 0.55      | $mol \ mol^{-1}$         |
|                      | Stationary combustion | $c_{ m sc}$  | 0.30  | 0.27 | 0.33 | 0.26 | 0.36 | 0.33      | $\mod \mathrm{mol}^{-1}$ |

et al., 2025). For the purposes of sectoral attribution, characteristic molar emission ratios of the road transport and stationary combustion sectors were calculated from the inventory for the three species ratios  $CO/CO_2$ ,  $NO_x/CO_2$ , and  $NO_x/CO$ . The characteristic molar ratios are spatial averages weighted by the total mass of emissions for each inventory grid cell. The ratios are summarised in Table 4 for the 4 x 4 km<sup>2</sup> area surrounding the tower as well as separate ratios per cardinal direction, as well as for the entire city of Zurich. While the full campaign-averaged footprint exceeds the limits of this 4 x 4 km<sup>2</sup> area, it contains the majority of the long-term flux footprint and the full city ratios do not deviate significantly from those found within the area.

# 2.5 Linear Mixing Model



Using simultaneous measurements of  $F_{\text{CO}_2}$ ,  $F_{\text{CO}}$ , and  $F_{\text{NO}_x}$ , along with expected reference flux ratios per species pair and source sector (Table 4) a linear mixing model is proposed to attribute measured (total,  $F_{x,\text{tot}}$ ) fluxes to biospheric ( $F_{x,\text{bio}}$ ) and combustion origin, with combustion fluxes attributed to either road transport ( $F_{x,\text{rt}}$ ) or stationary combustion ( $F_{x,\text{sc}}$ ) source categories. Here  $F_{x,\text{bio}}$  refers specifically to the net fluxes, i.e. biospheric sources (human, soil and plant respiration)

minus biospheric sinks (photosynthetic uptake). Biogenic fuel sources such as the portion of biofuel in vehicle fuel mixture is attributed to the combustive  $F_{x,\rm rt}$ .

We assume that for all three species all sources and sinks within the total flux-tower source area can be attributed to either the biospheric, road transport, or stationary combustion categories:

$$F_{\text{CO}_2,\text{tot}} = F_{\text{CO}_2,\text{bio}} + F_{\text{CO}_2,\text{rt}} + F_{\text{CO}_2,\text{sc}}.$$
 (1)

Secondly, we assume no biospheric sources or sinks of CO or NO<sub>x</sub> such that:

$$F_{\text{CO,tot}} = F_{\text{CO,rt}} + F_{\text{CO,sc}},\tag{2}$$

and

$$F_{NO_{x},tot} = F_{NO_{x},rt} + F_{NO_{x},sc}.$$
(3)

From the inventory, we determine four independent ratios as described in Table 4:

$$a_{\rm rt} = \frac{F_{\rm CO,rt}}{F_{\rm CO,rt}},\tag{4}$$

$$a_{\rm sc} = \frac{F_{\rm CO,sc}}{F_{\rm CO_2,sc}},\tag{5}$$

$$b_{\rm rt} = \frac{F_{\rm NO_x, rt}}{F_{\rm CO_2, rt}},\tag{6}$$

and

$$b_{\rm sc} = \frac{F_{\rm NO_x, sc}}{F_{\rm CO_2, sc}}.$$
 (7)

Two additional ratios are defined dependent on Eqs. 4 to 7:

$$c_{\rm rt} = \frac{a_{\rm rt}}{b_{\rm rt}} = \frac{F_{\rm CO,rt}}{F_{\rm NO_x,rt}},\tag{8}$$

and

$$c_{\rm sc} = \frac{a_{\rm sc}}{b_{\rm sc}} = \frac{F_{\rm CO,sc}}{F_{\rm NO,sc}}$$
. (9)

Using the budgets of Eqs. 1 to 3 and inventory ratios from 4 to 9 the total fluxes of  $NO_x$  and CO may be partitioned. Using Eqs. 8 and 9 in Eq. 2:

$$F_{\text{CO,tot}} = c_{\text{rt}} F_{\text{NO_x,rt}} + c_{\text{sc}} F_{\text{NO_x,sc}}, \tag{10}$$

and using Eq. 3 to eliminate  $F_{NO_x,rt}$ :

$$F_{\text{CO,tot}} = c_{\text{rt}} F_{\text{NO}_{x},\text{tot}} + F_{\text{NO}_{x},\text{sc}}(c_{\text{sc}} - c_{\text{rt}}). \tag{11}$$

Solving for  $F_{NO_x,sc}$ :

$$F_{\text{NO}_{x},\text{sc}} = \frac{F_{\text{CO,tot}} - c_{\text{rt}} F_{\text{NO}_{x},\text{tot}}}{(c_{\text{sc}} - c_{\text{rt}})}.$$
(12)

Similarly

$$F_{\text{NO}_{x},\text{rt}} = \frac{F_{\text{CO},\text{tot}} - c_{\text{sc}} F_{\text{NO}_{x},\text{tot}}}{(c_{\text{rt}} - c_{\text{sc}})},\tag{13}$$

$$F_{\text{CO,sc}} = \frac{F_{\text{NO}_x,\text{tot}} - \frac{1}{c_{\text{rt}}} F_{\text{CO,tot}}}{\left(\frac{1}{c_{\text{sc}}} - \frac{1}{c_{\text{rt}}}\right)},$$
 (14)

and


$$F_{\text{CO,rt}} = \frac{F_{\text{NO}_{x},\text{tot}} - \frac{1}{c_{\text{sc}}} F_{\text{CO,tot}}}{\left(\frac{1}{c_{\text{rt}}} - \frac{1}{c_{\text{sc}}}\right)}.$$
(15)

Equations 12 to 15 can be directly solved using simultaneous measurement of  $F_{\rm CO}$  and  $F_{\rm NO_x}$  and thus total fluxes of CO and NO<sub>x</sub> may be partitioned to these source sectors. These partitioned fluxes can then be used further to partition observed  $F_{\rm CO_2,tot}$ . The  $F_{\rm CO_2,tot}$  budget from Eq. 1 may be rewritten as:

$$F_{\text{CO}_2,\text{tot}} = F_{\text{CO}_2,\text{bio}} + \frac{F_{\text{CO,rt}}}{a_{\text{rt}}} + \frac{F_{\text{CO,sc}}}{a_{\text{sc}}}$$

$$\tag{16}$$

and therefore

$$F_{\text{CO}_2,\text{bio}} = F_{\text{CO}_2,\text{tot}} - \frac{F_{\text{CO},\text{rt}}}{a_{\text{rt}}} - \frac{F_{\text{CO},\text{sc}}}{a_{\text{sc}}}.$$
(17)

Following a similar approach as above the non-biospheric CO<sub>2</sub> can be partitioned in two component categories:

$$F_{\text{CO}_2,\text{sc}} = \frac{F_{\text{CO},\text{tot}} - a_{\text{rt}} (F_{\text{CO}_2,\text{tot}} - F_{\text{CO}_2,\text{bio}})}{(a_{\text{sc}} - a_{\text{rt}})}$$
 (18)

and

$$F_{\rm CO_2,rt} = \frac{F_{\rm CO,tot} - a_{\rm sc}(F_{\rm CO_2,tot} - F_{\rm CO_2,bio})}{(a_{\rm rt} - a_{\rm sc})}.$$
(19)

## 3 Results and Discussion

#### 3.1 Fluxes of Individual Species and their Correlation

Measured diurnal and seasonal patterns are shown for each species in Fig. 4 and summarised in Table 5. The median 24-hr  $F_{\text{CO}_2}$  in Aug-Oct was 7.4 µmol m<sup>-2</sup> s<sup>-1</sup> compared to 10.9 µmol m<sup>-2</sup> s<sup>-1</sup> in Nov-Mar with pronounced diurnal variability.

Table 5. Measured average diurnal fluxes and inter-quartile ranges for the fluxes shown in Fig. 4 separated by season.

|                            | Aug-Oct |      |            | Nov-Mar |      |            |                                                   |  |
|----------------------------|---------|------|------------|---------|------|------------|---------------------------------------------------|--|
| Species                    | Median  | Mean | IQR        | Median  | Mean | IQR        | Units                                             |  |
| $CO_2$                     | 7.4     | 10.7 | 3.0 - 15.3 | 10.9    | 16.9 | 5.2 - 22.3 | $\mu \text{mol m}^{-2} \text{ s}^{-1}$            |  |
| CO                         | 28      | 37   | 9 - 55     | 30      | 43   | 12 - 59    | $\mathrm{nmol}\ \mathrm{m}^{-2}\ \mathrm{s}^{-1}$ |  |
| $\mathrm{NO}_{\mathrm{x}}$ | 13      | 20   | 4 - 31     | 14      | 24   | 6 - 34     | $\mathrm{nmol}\ \mathrm{m}^{-2}\ \mathrm{s}^{-1}$ |  |
| $\mathrm{CH}_4$            | 13      | 20   | 1 - 31     | 13      | 24   | 3 - 31     | $\mathrm{nmol}\ \mathrm{m}^{-2}\ \mathrm{s}^{-1}$ |  |
| $N_2O$                     | 0.5     | 0.9  | 0.1 - 1.1  | 0.5     | 0.8  | 0.2 - 1.0  | $\rm nmol~m^{-2}~s^{-1}$                          |  |

 $F_{\rm CO_2}$  for both seasons begins to peak in the morning hours of 5-6 UTC (7-8 local time in the summer and 6-7 local time in the winter). Median nocturnal  $F_{\rm CO_2}$  is similar between seasons, varying by < 0.5  $\mu$ mol m<sup>-2</sup> s<sup>-1</sup> between 22 and 4 UTC. During the day  $F_{\rm CO_2}$  is on average higher in the winter and the median winter flux is 1.5 times greater than the median summer flux. Fluxes are more skewed in the winter where the mean  $F_{\rm CO_2}$  is 1.6 times the median, compared to the summer in which the mean is 1.4 times greater than the median. The largest median  ${\rm CO_2}$  fluxes come from the direction of the city centre to the SE (9.42  $\mu$ mol m<sup>-2</sup> s<sup>-1</sup>) while the smallest median fluxes (5.50  $\mu$ mol m<sup>-2</sup> s<sup>-1</sup>) come from the more vegetated SW (not shown). In contrast, there is much less seasonal variation in the CO fluxes, with median (mean)  $F_{\rm CO}$  of 30 (43) nmol m<sup>-2</sup> s<sup>-1</sup> in the winter compared to 28 (37) nmol m<sup>-2</sup> s<sup>-1</sup> in the summer, a median winter enhancement of 8%. A Welch's *t*-test indicates significant seasonal difference at p < 0.01. The clear one hour UTC offset in diurnal profiles between the seasons indicates that CO fluxes are dominated by anthropogenic rhythms, such as the work day and commuting hours, rather than responses to environmental conditions such as heating degree days. Nocturnal  $F_{\rm CO}$  is on average 2 times greater in the winter compared to the summer. In the winter, average CO fluxes were greatest from the SE (44 nmol m<sup>-2</sup> s<sup>-1</sup>), and in summer approximately even between the SE and NE (34 and 35 nmol m<sup>-2</sup> s<sup>-1</sup>, respectively). We expect the enhanced nocturnal CO from the city




core to be associated with residential heating.

Median hourly  $\mathrm{NO_x}$  fluxes exhibit a bimodal diurnal course. While the average magnitudes are similar between seasons in the afternoon mode, they are about 1.25 times greater in the winter compared to the summer during the morning mode. The median (mean)  $F_{\mathrm{NO_x}}$  was 14 (24) nmol m<sup>-2</sup> s<sup>-1</sup> in winter and 13 (20) nmol m<sup>-2</sup> s<sup>-1</sup> in the summer, a winter enhancement of 8% (20%). The seasons are significantly different at p 

Figure 4. Diurnal and seasonal pattern of fluxes of  $CO_2$ , CO,  $NO_x$ ,  $CH_4$  and  $N_2O$  measured by the CPEC. Weekdays (Monday to Friday) are shown as solid lines while weekends and holidays (Saturday, Sunday, and local holidays) are shown as dotted lines. The lines indicate the hourly medians and the shaded band the inter-quartile ( $P_{25}$  to  $P_{75}$ ) range. Note that the inter-quartile range is only shown for weekdays. Fluxes are divided by season, where red is August to October 2022 (inclusive) and blue is November 2022 to March 2023 (inclusive). In the sixth panel, air temperature ( $T_a$ ) as measured at the tower is shown.

are on average higher than daytime fluxes. The median winter flux is only 1% greater compared to summer: 13.5 vs. 13.3 nmol m<sup>-2</sup> s<sup>-1</sup>, and the mean winter flux is 18% greater: 23.9 vs. 20.3 nmol m<sup>-2</sup> s<sup>-1</sup>. Nevertheless the seasons are statistically significantly different at p < 0.01. N<sub>2</sub>O is the only species for which winter fluxes were observed to be smaller on average than summer fluxes, the median being 5% lower (0.49 to 0.52 nmol m<sup>-2</sup> s<sup>-1</sup>) and the mean 7% lower (0.83 to 0.89 nmol m<sup>-2</sup> s<sup>-1</sup>). However, a Welch's t-test indicates no statistically significant difference in mean  $F_{\rm N_2O}$  between the seasons (p = 0.34). Similarly to CH<sub>4</sub>, the nocturnal  $F_{\rm N_2O}$  was higher in the summer compared to the winter whereas the daytime  $F_{\rm N_2O}$  was lower, a ratio of 1.4 compared to 0.9 on average.


Significant weekday/weekend differences in fluxes (p < 0.01) are observed in all species except  ${\rm CH_4}$ . For  $F_{\rm CO_2}$  the difference is greater in the summer (1.9  $\mu$ mol m $^{-2}$  s $^{-1}$ ) compared to winter (0.4  $\mu$ mol m $^{-2}$  s $^{-1}$ ). The diurnal pattern also changes with season: in the warmer months the weekend  $F_{\rm CO_2}$  is flatter through the day, missing the morning mode of 5 - 7 UTC. In

**Table 6.** Summary of species fluxes. Results are split by season (Aug-Oct and Nov-Mar) and given for the median day (based on hourly medians) and mean day (based on hourly means). Results are presented as molar fluxes in mol m<sup>-2</sup> yr<sup>-1</sup> and mass fluxes in Mg km<sup>-2</sup> yr<sup>-1</sup>. For mass fluxes of  $CH_4$  and  $N_2O$  the  $CO_2$  equivalent (CO2e) mass flux is also given using 100-yr global warming potential factors of 28 ( $CH_4$ ) and 273 ( $N_2O$ ) (Forster et al., 2021).

|              |                        | Summer                   | (Aug-Oct)                 | Winter (                               | Nov-Mar)                  |
|--------------|------------------------|--------------------------|---------------------------|----------------------------------------|---------------------------|
|              |                        | Molar flux               | Mass flux                 | Molar flux                             | Mass flux                 |
|              |                        | $\bmod \ m^-2 \ yr^-1$   | $\rm Mg~km^-2~yr^-1$      | mol m <sup>-</sup> 2 yr <sup>-</sup> 1 | $\rm Mg~km^-2~yr^-1$      |
|              | $CO_2$                 | 357.0                    | 15 711.7                  | 501.24                                 | 22 059.8                  |
|              | CO                     | 1.06                     | 29.81                     | 1.30                                   | 36.44                     |
| Mean         | $NO_x$                 | 0.61                     | 28.18                     | 0.71                                   | 32.78                     |
|              | $\mathrm{CH}_4$        | 0.72                     | 10.82 (302.8 CO2e)        | 0.76                                   | 11.43 (320.0 CO2e)        |
|              | $N_2O$                 | 0.032                    | 1.43 (390.8 CO2e)         | 0.026                                  | 1.17 (318.1 CO2e)         |
|              | $\mathrm{CO}_2$        | 240.83 (104.49 - 493.69) | 10 599.0 (4 599 - 21 727) | 355.13 (172.91 - 674.63)               | 15 629.2 (7 610 - 29 690) |
|              | CO                     | 0.80 (0.36 - 1.46)       | 22.56 (10.1 - 40.9)       | 0.93 (0.46 - 1.61)                     | 26.19 (12.9 - 45.2)       |
|              | $NO_x$                 | 0.45 (0.15 - 0.86)       | 20.66 (7.2 - 39.6)        | 0.52 (0.16 - 0.95)                     | 23.70 (9.8 - 43.7)        |
| Median (IQR) | $\mathrm{CH}_4$        | 0.48 (0.07 - 1.02)       | 7.23 (1.0 - 15.3)         | 0.45 (0.11 - 0.98)                     | 6.83 (1.7 - 14.7)         |
|              | CH <sub>4</sub> (CO2e) |                          | 202.6 (28.1 - 428.8)      |                                        | 191.2 (46.6 - 411.9)      |
|              | $N_2O$                 | 0.018 (0.003 - 0.037)    | 0.81 (0.16 - 1.65)        | 0.015 (0.005 - 0.028)                  | 0.66 (0.23 - 1.26)        |
|              | $N_2O$ (CO2e)          |                          | 221.5 (43.0 - 449.5)      |                                        | 181.2 (62.3 - 345.2)      |

the colder months there remains a morning mode on the weekends, though it is distinctly shorter than that observed during weekdays. Further, there is an afternoon mode at 18 UTC that is only observed during the weekends. This trend extends to measurements of  $F_{\rm CO}$  as well as  $F_{\rm NO_x}$ : while weekend fluxes of these species are significantly lower in both seasons, an enhanced weekend winter mode at 18 UTC is observed for both species. The most pronounced weekday/weekend differences were observed for  ${\rm NO_x}$ , where median weekday fluxes were more than double median weekend fluxes (17 vs. 7 nmol m<sup>-2</sup> s<sup>-1</sup>). A small (< 0.1 nmol m<sup>-2</sup> s<sup>-1</sup>) but statistically significant weekday/weekend difference was found in both summer and winter for  $F_{\rm N_2O}$ . No statistically significant seasonal difference was found in either season for  $F_{\rm CH_4}$ 




The hourly mean and median fluxes were integrated and the results are summarised in Table 6. For greenhouse gases  $CH_4$  and  $N_2O$ , the  $CO_2$  equivalent mass flux is also given using the 100-year global warming potential according to Forster et al. (2021). Based on the mean day, the greenhouse gas emissions within the tower flux footprint consist of 95.8%  $CO_2$ , 1.8%  $CH_4$ , and 2.4%  $N_2O$  in the summer, and 97.2%  $CO_2$ , 1.4%  $CH_4$ , and 1.4%  $N_2O$  in the winter.

The correlation (Pearson's correlation coefficient r) between the fluxes of all species pairs on a 30-min scale is shown in Fig. 5. In the summer months (red, lower left) the highest correlation is seen between  $F_{\rm NO_x}$  and  $F_{\rm CO}$  at 0.72. Correlations at or above 0.6 were also observed between  $F_{\rm CO_2}$  and  $F_{\rm CO_3}$ , as well as between  $F_{\rm CO_2}$  and  $F_{\rm CH_4}$ . The lowest correlations ( $\leq$  0.3) were observed between  $F_{\rm NO_2}$  and  $F_{\rm CO_3}$ , as well as  $F_{\rm NO_3}$ , and  $F_{\rm CH_4}$ . In the winter months (blue, upper right),

**Figure 5.** Correlation matrix of 30-min fluxes of measured species. The lower-left half is the summer months (Aug-Oct) and upper-right half is the winter months (Nov-Mar).

correlations between nearly all species pairs increase, with the exception of  $F_{\rm CO_2}$  and  $F_{\rm CH_4}$  which decreases from 0.60 to 0.48. Correlations between  $F_{\rm CO}$  and  $F_{\rm NO_x}$  as well as  $F_{\rm CO_2}$  and  $F_{\rm N_{2O}}$  remain essentially unchanged (0.01 difference between seasons). Co-emitted species CO and  ${\rm NO_x}$  are selected as tracers for ffCO<sub>2</sub> in the following sections on the basis of their larger median fluxes and strong correlations in the winter.

## 3.2 Ratios of Species Fluxes





Flux ratios of  $NO_x$  and CO to  $CO_2$  were calculated along with the emission ratios for the largest emission sectors from the city emission inventory. In Fig. 6 the diurnal and seasonal ratios are shown for  $F_{NO_x}/F_{CO_2}$  (top) and  $F_{CO}/F_{CO_2}$  (bottom). The emission ratios determined from the inventory are shown as shaded bands using the range of values (wind-sector dependent) described in Table 4. Also shown as reference is the biospheric flux of human respiration which in both cases is 0.

As has been noted in other studies (Matthews and Schume, 2022; Helfter et al., 2016), tall-tower EC is more susceptible to large storage fluxes induced by boundary-layer dynamics that may be decoupled from actual surface emissions within a given 30-minute period. Emissions that accumulate within the nocturnal boundary layer and below the height of the EC system can register as anomalously large surface fluxes during the morning thermal mixing and breakup of the nocturnal boundary layer. The single-point storage correction employed here is rudimentary and may underestimate the true storage flux by over 50% (Finnigan, 2006). However, even in the case of significantly underestimated storage fluxes, flux ratios should be unaffected and remain representative of the stored emission sources, which simplifies sectoral attribution. The steep increases in morning fluxes seen in CO<sub>2</sub>, CO, and NO<sub>x</sub> in Fig. 4, e.g., do not correspond to similarly steep rises in their emission ratios in Fig. 6.

In both seasons and for both ratios, the diurnal minima are observed during the night. The observed  $F_{NO_x}/F_{CO_2}$  nocturnal ratios are higher in winter than summer with median (mean) ratios of 0.89 (1.51) mmol mol<sup>-1</sup> in the winter and 0.69 (0.76)

Figure 6. Diurnal ratios of  $F_{NO_x}$  to  $F_{CO_2}$  (top) and  $F_{CO}$  to  $F_{CO_2}$  (bottom) separated by season. The range of inventory emission ratios from Table 4 is shaded.

mmol mol<sup>-1</sup> in the summer. Between 0 and 2 UTC in the summer, the median ratios are within 10% of the expected range of stationary combustion ratios. Outside these times, and for all hours during the winter, the median observed ratios are above the range expected for pure stationary combustion. The diurnal maxima in  $F_{\rm NO_x}/F_{\rm CO_2}$  in both seasons are observed in the afternoon between 13 and 14 UTC. In the winter, the measured median  $F_{\rm NO_x}/F_{\rm CO_2}$  is < 0.1 mmol mol<sup>-1</sup> from the inventory reference ratio for road transport between 13 and 15 UTC before gradually reducing through the afternoon and evening. These measurements show very good agreement between measured  $F_{\rm NO_x}/F_{\rm CO_2}$  and expected inventory  ${\rm NO_x}/{\rm CO_2}$  molar emission ratios through the winter and a clear diurnal cycle that suggests observed ratios are dominated by stationary combustion through the night, dominated by road transport in the afternoon, and a mixture of the two during other times of day. This agreement is strong evidence for both the accuracy of the emission inventory as well as the possibility for emission ratios to reduce the artifacts of storage fluxes from single-point EC measurement towers. Observed nocturnal ratios are systematically higher in winter than summer. This may be attributed to biospheric respiration of  ${\rm CO_2}$ , which is higher in the summer and produces  $F_{\rm NO_x}/F_{\rm CO_2}$  of 0. In both seasons biospheric  ${\rm CO_2}$  from human respiration is also expected to affect measured ratios.




 $F_{\mathrm{NO_x}}/F_{\mathrm{CO_2}}$  ratios observed through the summer follow a similar diurnal pattern, but with ratios that exceed the expected upper ratio from road transport between 11 and 16 UTC. Between these hours, the median measured ratio is between 0.5 and 1.9 mmol  $\mathrm{mol^{-1}}$  greater than the expected ratio for road transport. This seasonal difference is a strong indication of photosynthetic activity within the footprint causing a decrease in the absolute  $F_{\mathrm{CO_2}}$ , and therefore a systematic increase in  $F_{\mathrm{NO_x}}/F_{\mathrm{CO_2}}$ . The median summer ratio is 1.5 mmol  $\mathrm{mol^{-1}}$  compared to 1.2 mmol  $\mathrm{mol^{-1}}$  in the winter and the mean ratios are 2.0 to 1.4 mmol

 $m mol^{-1}$ , respectively. Another seasonal difference is seen in the morning hours before 11 UTC where the summer  $F_{
m NO_x}/F_{
m CO_2}$  exceeds the winter and tends on average towards the road transport ratio. This may be attributed to the use of heating during the colder winter months causing a more mixed signal, whereas in the summer months, in the absence of significant heating, road transport is the more dominant signal.

Similar patterns are observed in the bottom panel of Fig. 6 for the  $F_{\rm CO}/F_{\rm CO_2}$  ratios. Nocturnal ratios in the summer from 22 to 4 UTC agree well with the inventory stationary combustion ratio, on average < 0.3 mmol mol<sup>-1</sup> from the median expected value. In contrast to the  $F_{\rm NO_x}/F_{\rm CO_2}$  ratios, the  $F_{\rm CO}/F_{\rm CO_2}$  remain elevated well above this ratio from 18 to 21 in both seasons and in the winter remain outside the expected range for stationary combustion through 23 UTC. Median nocturnal ratios in the winter are 35% greater than the summer (1.9 to 1.4 mmol mol<sup>-1</sup>). During the winter daytime, a similar increase towards the expected road transport ratios is observed and the median flux ratio falls within this range from 12 to 18 UTC, with a mixture between the two source categories during other hours. Again in the summer there is a systematic exceedance of the upper road transport ratios during the afternoon. From 12 to 17 UTC, the median ratio is 0.7 to 1.8 mmol mol<sup>-1</sup> greater than the upper road transport reference ratio, and the median summer daytime ratio is 42% greater (4.4 to 3.1 mmol mol<sup>-1</sup>) than the winter. Again, the observed median hourly  $F_{\rm CO}/F_{\rm CO_2}$  in the winter are well bounded by the expected values determined from the emissions inventory whereas a systematic exceedance in the summer afternoon suggests biospheric influence.

Both panels in Fig. 6 suggest that reported inventory ratios are reasonable and the emissions within the tower footprint may be well characterised by a combination of stationary combustion and road transport during the winter. Nevertheless, there remain significant differences in the diurnal courses between the  $F_{\rm NO_x}/F_{\rm CO_2}$  and  $F_{\rm CO}/F_{\rm CO_2}$  ratios. Where the summer  $F_{\rm NO_x}/F_{\rm CO_2}$  afternoon exceedance has one mode and peaks at 14 UTC, the  $F_{\rm CO}/F_{\rm CO_2}$  arguably has two modes whose valley occurs during 13 and 14 UTC, contradicting the trend of the  ${\rm NO_x}$  ratios. The evening (18 to 23 UTC) winter  $F_{\rm CO}/F_{\rm CO_2}$  ratios remain elevated towards the expected road transport ratio, suggesting a CO source in addition to road transport and stationary combustion. One possible explanation is wood combustion, which has a very high  ${\rm CO/CO_2}$  molar ratio of, e.g., 50 to 110 mmol mol<sup>-1</sup> depending on appliance and wood species (Evtyugina et al., 2014), and even a small number of wood-fired heating systems could therefore elevate the observed  $F_{\rm CO}/F_{\rm CO_2}$ .

The diurnal and seasonal flux ratios for  $NO_x/CO$  are shown in Fig. 7. In contrast to Fig. 6, there is no immediately clear seasonal variability in  $F_{NO_x}/F_{CO}$  and the median winter and summer fluxes vary by 0.014 mol mol<sup>-1</sup> (0.485 mol mol<sup>-1</sup> in the winter, 0.471 mol mol<sup>-1</sup> in summer). While the mean summer ratio is double the mean winter ratio (0.852 against 0.425 mol mol<sup>-1</sup>), this may be attributed to outliers when absolute flux magnitudes of both species become very small. A Welch's t-test indicates no statistically significant difference in means between the seasons (p = 0.49). The small absolute flux magnitudes contribute to considerable variability through much of the morning hours though in winter the median ratios from 0 to 3 UTC and 19 to 22 UTC vary by < 0.05 mol mol<sup>-1</sup>. From 3 to 9 UTC and again 13 to 14 UTC the median ratios in both seasons exceed the upper expected ratio for road transport. These measured ratios overwhelmingly come from the NE wind sector where the median winter ratio is 1.4 to 1.9 times greater than from the other directions (median 0.73 mol mol<sup>-1</sup> vs. 0.46 (SE), 0.39 (SW), and 0.51 (NW) mol mol<sup>-1</sup>). This enhancement from the NE is also seen in summer where it is 1.2 to 1.6 times greater than other directions and is driven by a similar pattern in enhanced  $NO_x$  fluxes (rather than lower CO fluxes). This may

Figure 7. Diurnal and seasonal ratios of  $F_{NO_x}$  to  $F_{CO}$ . Inventory emission ratios from Table 4 are shaded.





indicate the influence of a source sector with much higher  $NO_x$  to CO emission ratios to the NE. Emission ratios from the NW are also higher during rush hours than the highest expected road transport ratio and the expected ratios shown in Table 4 only characterise the SW and SE wind directions well.

Despite this directional variability, measured winter  $F_{\rm NO_x}/F_{\rm CO}$  ratios broadly show higher measured ratios around 6-7 UTC and 13-14 UTC and lower ratios through the evening and night, though this pattern is muted by strong dependence on wind direction. The measured summer  $F_{\rm NO_x}/F_{\rm CO}$  show a similar afternoon enhancement, with a longer morning enhancement (3-8 UTC) driven by a greater prevalence of NE winds. The lower ratios during afternoon rush hours compared to morning rush hours may be due to higher CO emissions from vehicle cold-starts which can lead to larger CO emissions from similar traffic counts (Jayaratne et al., 2021). The lack of significant seasonal variability in the  $F_{\rm NO_x}/F_{\rm CO}$  ratio suggests that the daytime seasonal variability in  $F_{\rm NO_x}/F_{\rm CO_2}$  and  $F_{\rm CO}/F_{\rm CO_2}$  is due to seasonal changes in  $F_{\rm CO_2}$ .

In Fig. 8 the flux ratios are directly compared. In Fig. 8-A a 2d histogram of measured  $F_{\rm NO_x}/F_{\rm CO_2}$  and  $F_{\rm CO}/F_{\rm CO_2}$  for the full campaign and the area bounded by the area-averaged inventory emission ratios is shown. While the general shape of the 2d histogram of the measured fluxes is described by the inventory ratios, a large number of observations fall outside this boundary. Notably, while the peak of the distribution falls within the expected range for stationary combustion for  $F_{\rm CO}/F_{\rm CO_2}$ , the  $F_{\rm NO_x}/F_{\rm CO_2}$  peak falls outside the expected range.

Histograms of seasonal fluxes are shown in Fig. 8-B along with the range expected from the emissions inventory. Here again the seasonal variability in  $F_{\rm NO_x}/F_{\rm CO_2}$  and  $F_{\rm NO_x}/F_{\rm CO}$  is shown and the lack of significant seasonal variability in  $F_{\rm NO_x}/F_{\rm CO}$  is made clear. The dotted horizontal lines indicating the inventory ranges show that for each species ratio the majority of fluxes are well captured by the inventory and the sectors of road transport and stationary combustion in both summer and winter.

In the winter, however, the peaks of the distributions for  $F_{\rm NO_x}/F_{\rm CO_2}$  and  $F_{\rm CO}/F_{\rm CO_2}$  are significantly higher, indicating that in the winter these sectors are better able to characterise the measured flux ratios. In the summer, the distribution peaks decrease and a greater portion of measured fluxes occur outside the inventory bounds. That this difference is not repeated in the  ${\rm NO_x}$  to  ${\rm CO}$  ratios is a strong indication that biospheric activity is being measured.

Figure 8. A - Heatmap of measured  $F_{\rm NO_x}/F_{\rm CO_2}$  and  $F_{\rm CO}/F_{\rm CO_2}$  ratios. Emission ratios from Table 4 are labelled and the area between is bounded by the dashed lines. B - Histograms of measured  $F_{\rm NO_x}/F_{\rm CO_2}$ ,  $F_{\rm CO}/F_{\rm CO_2}$ , and  $F_{\rm NO_x}/F_{\rm CO}$  split by season. Inventory ratio ranges are shown as horizontal dotted lines.

A non-negligible amount of winter observations occur outside the range predicted by the inventory. These may be attributed to a few reasons: Firstly, the selected inventory ratio ranges are comprised of a small number of spatially-averaged values with the intention of determining a characteristic ratio range for each source category. While these are weighted to the total mass of yearly emissions, variations in individual half-hour flux footprints may still sample areas outside these ratios. When the full range of ratios for each grid cell surrounding the tower is considered, the range of plausible values is greatly expanded: For  $CO/CO_2$  the range is 0.13 to 89.17 mmol  $mol^{-1}$ . For  $NO_x/CO_2$  0.19 to 3.78 mmol  $mol^{-1}$ . For  $NO_x/CO$  0.016 to 4.58 mol  $mol^{-1}$ . While the extremes of the ranges represent cells with small contributions to total emissions, values outside the spatially-averaged mean may still be plausible depending on the individual 30-minute footprint. The overlap of the chosen inventory ratios with the histogram modes of Fig. 8-B indicates the values from Table 4 are acceptable characteristic ranges for the measured ratios despite not describing all measurements. Nevertheless, from Fig. 8-A it seems that nocturnal  $F_{NO_x}$  measurements (Fig. 8 at  $F_{CO}/F_{CO_2} \approx 1.3$ ,  $F_{NO_x}/F_{CO_2} \approx 0.6$ ) especially are higher than predicted by the inventory.




Secondly, depending on the footprint, the contribution from sectors other than road transport and stationary combustion may impact measured ratios. While these two sectors comprise 91% of  $\rm CO_2$  from combustion in the 4 x 4 km<sup>-2</sup> area surrounding the tower according to the emission inventory, they comprise 81% of  $\rm CO$  and 84% of  $\rm NO_x$ . Some sources that may also contribute to the measured flux fall within the same ratio range. E.g., 7% of  $\rm SW$   $F_{\rm CO}$  may be expected from industry where the expected  $\rm CO/CO_2$  ratio of 2.48 mmol  $\rm mol^{-1}$  falls between the expected stationary combustion and road transport values. In contrast, 13% of  $F_{\rm NO_x}$  from the NE is expected from public power, with a characteristic  $\rm NO_x/CO$  ratio of 3.82 mol  $\rm mol^{-1}$ , well outside the  $\rm NO_x/CO$  ratios of Table 4.

Finally, in the case of  $F_{NO_x}/F_{CO_2}$  and  $F_{CO}/F_{CO_2}$ , depending on the vegetation within the footprint, there may still be photosynthetic  $CO_2$  uptake even through the winter (Wu et al., 2022) which produces higher than expected ratios during day, or enhanced  $CO_2$  fluxes caused by respiration which may produce lower than expected ratios during night.

## 3.3 Flux Partitioning and Sectoral Attribution



The linear mixing model described in Section 2.5 was applied and the decomposed fluxes of CO,  $NO_x$ , and  $CO_2$  are shown in Figure 9. Model input parameters for a and b were chosen from the full 4 x 4 km<sup>2</sup> footprint area of Table 4 and were kept constant across times and seasons.

Figure 9-A shows the source sector decomposition of  $F_{\rm CO}$ . In summer and winter the majority of the total flux (57 and 59% respectively) is attributed to road transport, with the stationary combustion portion comprising 43% of the median flux in summer and 41% of the median flux in winter. In both seasons a bimodal distribution in  $F_{\rm CO,rt}$  captures local rush hours. In the summer, the morning hours 0-6 UTC are attributed entirely to road transport, while in winter a higher median observed flux from 0-1 UTC is partitioned towards a small positive contribution by stationary combustion ( $F_{\rm CO,sc}$ ). During the peak of the morning rush hour in both seasons the median modelled  $F_{\rm CO,rt}$  exceeds the measured  $F_{\rm CO,tot}$  (by 1.24 times at 6 UTC in summer and 1.12 times at 7 UTC in winter), resulting in unrealistic negative  $F_{\rm CO,sc}$  at these times as  $F_{\rm CO,sc}$  and  $F_{\rm CO,rt}$  must sum to  $F_{\rm CO,tot}$  for each half-hour period. Through the course of the day,  $F_{\rm CO,sc}$  generally increases and in both seasons reaches a daily maximum after the afternoon road transport mode. In the summer,  $F_{\rm CO,tot}$  remains partitioned approximately equally between  $F_{\rm CO,rt}$  and  $F_{\rm CO,sc}$  until 23 UTC, while in winter the higher  $F_{\rm CO,tot}$  in the evening hours remains largely attributed to  $F_{\rm CO,sc}$  as  $F_{\rm CO,rt}$  declines following the afternoon rush-hour maximum. The highest proportion of  $F_{\rm CO,sc}$  is found in the evening hours in the winter (19-23 UTC) when it comprises 50-70% of  $F_{\rm CO,tot}$ .

Similar patterns are observed for the sector decomposition of  $F_{\rm NO_x}$  (Fig. 9-B). The  $F_{\rm NO_x,tot}$  is overwhelmingly (91%) partitioned to  $F_{\rm NO_x,rt}$ , with  $F_{\rm NO_x,sc}$  comprising only 9% of the total flux in both seasons. The median hourly  $F_{\rm NO_x,sc}$  never exceeds the median hourly  $F_{\rm NO_x,rt}$ , reaching a maximum of 40% to 60% of  $F_{\rm NO_x,rt}$  in the evening winter hours.

In Fig. 9-C the attribution of  $F_{\rm CO_2,tot}$  to  $F_{\rm CO_2,sc}$ ,  $F_{\rm CO_2,rt}$ , and the biospheric component  $F_{\rm CO_2,bio}$  is shown. The comparatively large nocturnal  $F_{\rm CO_2,tot}$  in the early morning hours is attributed largely to biospheric activity in both summer and winter with a larger summertime  $F_{\rm CO_2,bio}$  (5.5  $\mu$ mol m<sup>-2</sup> s<sup>-1</sup> average from 0 to 6 UTC) than wintertime  $F_{\rm CO_2,bio}$  (4.2  $\mu$ mol m<sup>-2</sup> s<sup>-1</sup>), both in terms of absolute flux and as compared to  $F_{\rm CO_2,rt}$  and  $F_{\rm CO_2,sc}$ . In both seasons there is a large daytime negative  $F_{\rm CO_2,bio}$ , starting from 8 UTC in summer and from 10 UTC in winter. In the summer, median hourly  $F_{\rm CO_2,bio}$  remains negative and between -7 to -10.5  $\mu$ mol m<sup>-2</sup> s<sup>-1</sup> until 20 UTC when it again becomes positive and ranges from 1.3 to 6.9  $\mu$ mol m<sup>-2</sup> s<sup>-1</sup> between 20 and 7 UTC. In the winter, the daytime negative flux is smaller: between -0.2 and and -7  $\mu$ mol m<sup>-2</sup> s<sup>-1</sup>. However, it remains negative through 22 UTC. Modelled CO<sub>2</sub> exchange of parks within the tower footprint shows daytime magnitudes of around -10  $\mu$ mol m<sup>-2</sup> s<sup>-1</sup> during the summer months but little exchange in the winter (0 ±1  $\mu$ mol m<sup>-2</sup> s<sup>-1</sup>) (Stagakis et al., 2024). The negative nocturnal  $F_{\rm CO_2,bio}$  is unrealistic, and stems from the unrealistically large  $F_{\rm CO_2,sc}$  partitioning, which from 17 to 22 UTC is 100% to 155% of the measured  $F_{\rm CO_2,tot}$ .

**Figure 9.** Modelled attribution of measured fluxes to source sector via the linear mixing model described in Section 2.5.  $A - F_{CO}$ ,  $B - F_{NO_x}$ ,  $C - F_{CO_2}$ . Median hourly course in UTC is shown, and the vertical bars indicate the inter-quartile range ( $P_{25}$  to  $P_{75}$ ) of the hourly bins. Differences in the diurnal course of measured fluxes from Fig. 4 may be attributed to the exclusion of the NE wind directions.

**Table 7.** Partitioning of fluxes compared to inventory for all combinations of input reference ratios for the full 8 month period. Inventory partitioning is given as a percentage of the total combustive flux of a species (% of total), i.e., considering all contributing source categories (see Table 3), as well as relative to each other (i.e., assuming the full flux may be attributed to one of the two modelled source categories). Model outputs are given as  $P_{25}$  -  $P_{75}$  ( $P_{50}$ ) using the distribution of all model outputs.

|                                | Inventory F | Reference |                  |                | Modelled Partitioning |                  |                  |  |  |
|--------------------------------|-------------|-----------|------------------|----------------|-----------------------|------------------|------------------|--|--|
| Species                        | % of total  | relative  | All              | SE             | SW                    | NW               | NE               |  |  |
| CO,sc                          | 38%         | 48%       | -58 - 9 (-16%)   | -23 - 26 (6%)  | 27 - 64 (45%)         | -13025 (-66%)    | -28695 (-163%)   |  |  |
| CO,rt                          | 42%         | 52%       | 78 - 146 (105%)  | 55 - 106 (76%) | 25 - 62 (43%)         | 118 - 227 (159%) | 192 - 388 (264%) |  |  |
| $NO_x$ ,sc                     | 30%         | 36%       | -35 - 5 (-9%)    | -15 - 17 (4%)  | 19 - 55 (34%)         | -6912 (-31%)     | -12540 (-70%)    |  |  |
| $\mathrm{NO}_{\mathrm{x}}$ ,rt | 54%         | 64%       | 83 - 126 (99%)   | 67 - 101 (81%) | 34 - 71 (55%)         | 104 - 162 (122%) | 135 - 221 (164%) |  |  |
| $\mathrm{CO}_2$ ,sc            | 67%         | 74%       | -157 - 23 (-44%) | -45 - 51 (12%) | 70 - 184 (122%)       | -30057 (-148%)   | -790250 (430%)   |  |  |
| $\mathrm{CO}_2$ ,rt            | 23%         | 26%       | 70 - 109 (85%)   | 37 - 60 (47%)  | 23 - 49 (37%)         | 89 - 144 (109%)  | 173 - 291 (215%) |  |  |
| $\mathrm{CO}_2$ ,bio           | -           | -         | 30122 (-70%)     | -6 - 95 (35%)  | -4521 (-37%)          | -84206 (-137%)   | -217434 (-307%)  |  |  |

While certain reasonable patterns may be discerned from these results, it is also clear that the model is sensitive to the reference inventory ratios and can produce improbable and impossible results. The ratios chosen here produce a partitioning that significantly underestimates the contribution from stationary combustion in winter. From the inventory, annual mean  $F_{\rm CO,rt}$  to  $F_{\rm CO,sc}$  should be nearly 1:1 (38% vs. 42% of total) rather than the approximately 2:1 shown here (66% vs. 30% of measured CO). Similarly,  $F_{\rm NO_x,rt}$  to  $F_{\rm NO_x,sc}$  is expected to be approximately 2:1 (54% vs. 30% of total) rather than the 10:1 (89% vs. 9% of measured NO<sub>x</sub>) model partitioning.  $F_{\rm CO_2,sc}$  is expected to be 3 times  $F_{\rm CO_2,rt}$  (67% vs. 23% of total) but in the model they are partitioned roughly 1:1 (54% vs. 50% of measured CO<sub>2</sub>). The mismatch in the observed stationary combustion ratios shown in Fig. 8-A is a likely source of poor partitioning of  $F_{\rm NO_x}$  into stationary combustion.

To test the sensitivity of the linear mixing model to the inventory reference ratios required as 0input, the model was re-run to consider all combinations of input reference ratios a (CO/CO<sub>2</sub>) and b (NO<sub>x</sub>/CO<sub>2</sub>) from Table 4 at increments of 0.05 mmol  $^{-1}$  (i.e., a model was run for  $a_{rt} = 3.00, 3.05, 3.10, ... 4.40$ , etc.), and the distribution of all model outputs (median and IQR) presented for the entire area as well as per wind direction sector. The results are summarised in Table 7, compared against the total inventory (i.e., Table 3) as well as relative to each other (i.e., assuming only two combustion source categories) and provide insight in to the sensitivity of the model to small changes in input reference ratios, as well as possible explanations for poor partitioning. There are distinct directional differences regarding the partitioning of observed fluxes into source categories. While the range of model outputs is reasonable to the SW, there is no combination of inputs from Table 4 that produce reasonable partitioning of fluxes from the NE. The NW is also poorly partitioned while the SE produces a more reasonable partitioning. As shown in Fig. 9, negative partitioning occurs where measured flux ratios exceed the limits predicted by the inventory reference ratios.

It is not immediately clear from the inventory why this is so as the SW is similar to the other directions, with road transport and stationary combustion accounting for 80% of  $\rm CO$ , 79% of  $\rm NO_x$ , and 92% of combustive  $\rm CO_2$ , with minority contributions from industry, offroad, and waste categories. To the NE the previously discussed contribution from public power may be causing both the higher observed fluxes from this direction, and subsequently poorer partitioning of said fluxes. Negative partitioning occurs where measured ratios exceed the limits defined by the linear mixing model. There are several implicit and explicit assumptions made in this model, violations of which could produce erroneous attribution:

A1. The emissions inventory is correct,






- A2. The combustion emission ratios are unchanging in time,
- A3. The combustion emission ratios are spatially homogeneous,
- 515 A4. The total flux of NO<sub>x</sub> and the total flux of CO contain only non-biospheric components, and
  - A5. Measured combustion fluxes are solely due to either road transport, stationary combustion, or a mixture of the two.

A1 means that we are assuming no errors with regards to spatial upscaling or distribution of aggregated emissions ratios within the city as well as perfect characterisation of the vehicle fleet and stationary combustion processes. Note that this does not require the total mass of yearly emissions to be accurate per se, but only that the per-sector emission ratios are well characterised. A2 assumes that the emission ratio for a given category, e.g., road transport, is static in time and changes neither with fleet composition nor time of day nor time of year nor with changing environmental conditions. The fleet composition may be assumed to be unchanging on average for the short duration covered by this campaign, though trends in fleet composition in Zurich are resulting in a long-term decrease in NO<sub>x</sub> from road transport. While the fleet composition on average may be unchanging, the different rhythms of traffic types (e.g., light-duty delivery vehicles vs. personal commuter cars) may lead to variation in the actual emission ratios in any half-hour. Further, the influence of engine cold starts may produce diurnal variation in emission ratios. Jayaratne et al. (2021) showed that similar traffic counts during morning and afternoon rush hours in a city produce larger CO concentrations in the afternoon due to high CO emissions during cold starts which are concentrated within the urban core. Further, cold start enhancement of CO<sub>2</sub>, CO, and NO<sub>x</sub> exhibits a non-linear dependence on ambient temperature (Weilenmann et al., 2009; Bielaczyc et al., 2011). A3 assumes that the emission ratio of a given category does not change with direction/source area. As discussed in Sections 2.4 and 3.1 there is spatial variation in the per-grid cell inventory ratios surrounding the tower, and spatial heterogeneity is a perennial characteristic of urban emissions. Together A2 and A3 allow each emission category and species ratio to be characterised by a single value for the extent of the campaign. A4 is unlikely to be significantly violated, yet there could be biogenic production of CO due to VOCs (Griffin et al., 2007). A5 is untrue for the city as a whole, but a reasonable simplification for the 4 x 4 km<sup>2</sup> area surrounding the tower. Though as discussed, the exceptions to A5 from certain sectors and directions could drive ratios well outside of the predicted range: enhanced  $F_{\rm NO}$ from the NE especially contributes to a greater attribution of the flux to road transport. Finally, it must be noted that the linear correction applied to the  $NO_x$  measurements may cause an overestimation of  $F_{NO_x}$  in some or all periods. This could lead to an over-attribution of net fluxes of all three species to road transport if the lower  $F_{NO_x}/F_\chi$  ratios of stationary combustion are not met, even if all other assumptions are met. An excessive correction would explain this over-attribution as well as the apparent elevated measurements of Fig. 8.

Although the linear mixing model produces expected results in many situations, exceptions to A2 and A5 in the complex and heterogenous urban environment may ultimately pose too great a challenge for the application of such a model with fixed emission factors over long periods of time and large flux footprints. In future work, there should be a determination of where and when the model fails using footprint modelling of individual flux averaging periods as well as a spatially and temporally resolved high-resolution emission inventory.

# 4 Summary and Conclusions








This work showcased eight months (August 2022 to March 2023) of continuous urban tall-tower eddy covariance measurements of  $CO_2$  and four co-emitted species: CO,  $NO_x$ ,  $CH_4$ , and  $N_2O$ . To our knowledge it is the first work to demonstrate simultaneous flux measurements of these species over an urban area. While the EC flux footprint does not cover the complete administrative boundary of the city of Zurich, it is clear from these measurements that the city is a net source of  $CO_2$ , CO,  $NO_x$ ,  $CH_4$ , and  $N_2O$ .

Considering the 100-year global warming potential of non-CO<sub>2</sub> greenhouse gases, the overall CO<sub>2</sub>-equivalent emissions within the tower footprint consist of 95.8% CO<sub>2</sub>, 1.8% CH<sub>4</sub>, and 2.4% N<sub>2</sub>O in summer, and 97.2% CO<sub>2</sub>, 1.4% CH<sub>4</sub>, and 1.4% N<sub>2</sub>O in winter. Median wintertime enhancement of  $F_{\rm CO_2}$  was largest at 1.47 times larger than summer (10.9 to 7.4 µmol m<sup>-2</sup> s<sup>-1</sup>) while seasonal enhancement of other species was smaller: 1.08 times larger for  $F_{\rm CO_2}$ ; 1.08 times larger for  $F_{\rm NO_x}$ ; 1.01 times larger for  $F_{\rm CH_4}$ ; and 0.95 times (smaller) for  $F_{\rm N_{2}O}$ . Correlation between  $F_{\rm CO_2}$  was highest with  $F_{\rm CO}$  in the summer and with  $F_{\rm CO}$  and  $F_{\rm NO_x}$  in the winter.

Observed flux ratios of the three most-emitted species ( ${\rm CO_2}$ ,  ${\rm CO}$ , and  ${\rm NO_x}$ ) were calculated and compared to characteristic spatially-averaged values from a city emission inventory within a 4 x 4 km² area centred on the measurement tower. Ratios of  $F_{\rm NO_x}/F_{\rm CO_2}$  and  $F_{\rm CO}/F_{\rm CO_2}$  in winter were generally well characterised by the stationary combustion and road transport categories of the emission inventory. Significant and systematic exceedance of the inventory ratios was observed in the summer afternoon hours, which was attributed to decreased net  $F_{\rm CO_2}$  due to photosynthetic uptake. Measured ratios of  $F_{\rm NO_x}/F_{\rm CO}$  were less well characterised by the inventory, with significant exceedance of the expected ratios coming from the NE. Nevertheless road traffic peaks seem well characterised by the measured  $F_{\rm NO_x}/F_{\rm CO}$  ratios. No statistically significant seasonal difference was observed in  $F_{\rm NO_x}/F_{\rm CO}$  ratios. Note that a significant bias correction was applied to raw measurements of NO<sub>x</sub> (see Sec. 2.3). This correction introduces a significant but unquantified uncertainty to individual flux measurement periods.

A linear mixing model for partitioning observed  $F_{\rm CO_2}$ ,  $F_{\rm CO}$ , and  $F_{\rm NO_x}$  into stationary combustion, road transport, and biospheric components was proposed and tested. Using characteristic spatially-averaged reference ratios for each category from the city emissions inventory, positive  $F_{\rm CO_2,bio}$  was found during the night and negative  $F_{\rm CO_2,bio}$  during the day in the warm months. In the colder months, positive  $F_{\rm CO_2,bio}$  was found during the morning hours and a reduced, but still negative,

 $F_{\mathrm{CO_2,bio}}$  was found from mid-day through midnight.  $F_{\mathrm{CO_2,rt}}$  and  $F_{\mathrm{CO_2,sc}}$  were partitioned approximately evenly: 50% and 54% of  $F_{\mathrm{CO_2,tot}}$ , respectively.  $F_{\mathrm{CO,rt}}$  was overestimated compared to the inventory, with 66% of  $F_{\mathrm{CO,tot}}$  attributed to  $F_{\mathrm{CO,rt}}$  and 30% to  $F_{\mathrm{CO,sc}}$ .  $F_{\mathrm{NO_x,rt}}$  was highly overestimated, with 89% of  $F_{\mathrm{NO_x,tot}}$  attributed to  $F_{\mathrm{NO_x,rt}}$  and 9% attributed to  $F_{\mathrm{NO_x,sc}}$ . Total partitioning of species was very sensitive to inventory reference inputs and did not perform equally well for all cardinal directions.



This work demonstrates the potential to partition  $\mathrm{CO}_2$  fluxes into different source and sink sectors using  $\mathrm{CO}$  and  $\mathrm{NO}_x$  as coemitted species. Measurement of co-emitted species provides important additional information on the probable source sector contribution to individual 30-minute fluxes, however in the complex and heterogenous urban environment this information is difficult to exploit on its own, without the use of a spatio-temporally-resolved emission inventory and individual flux footprint modelling. It also highlights some challenges of simultaneous measurement of fluxes of co-emitted species via tall-tower urban eddy covariance. In the next steps of this work, these measurements will be combined with modelled 30-minute flux footprints as well as a temporally resolved city emission inventory to gain a more detailed understanding of flux ratios and their variability, as well as further validation of the inventory itself.

Data availability. The raw data and processed fluxes used in this analysis are available from the ICOS Cities carbon portal https://citydata.icos-585 cp.eu/portal/. Author contributions. RH oversaw the CPEC measurement system, wrote the manuscript and created the figures, and performed the analysis. JH oversaw the CPEC installation, initial operation, and early analysis. SS installed and oversaw the OPEC measurements. BM produced the footprint and surface-cover analysis based on the model of NK. LC and DB created the emissions inventory. AK operated the flask sampler used for instrument comparison. AC conceived the flux partitioning model. All authors reviewed and contributed to the manuscript and contributed intellectually to the work.

Competing interests. The authors declare that they have no conflict of interest.



Acknowledgements. We thank Roland Vogt and Christian Feigenwinter from the University of Basel for installation and maintenance of the EC site. We thank Matthias Zeeman from the University of Freiburg for managing the data infrastructure. We thank Felix Baab and Dirk Redepenning, also from the University of Freiburg, for logistics and hardware for the EC site. We also thank Armund Sigmund from the University of Basel for his analysis of building distortion on the measured wind fields. The authors have received funding from ICOS Cities, a.k.a. the Pilot Applications in Urban Landscapes – Towards integrated city observatories for greenhouse gases (PAUL) project, from the European Union's Horizon 2020 research and innovation program under grant agreement no. 101037319. Financial support from ICOS Switzerland (ICOS-CH) Phase 3 (Swiss National Science Foundation, grant 20FI20\_198227) is also acknowledged.

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
