# Peer review of "Sectoral attribution of greenhouse gas and pollutant emissions using multi-species eddy covariance on a tall tower in Zurich, Switzerland"

_EGUsphere, 2025_

## Author Comment (AC1)

**Reviewer 1 - Martijn Pallandt**

The authors wish to thank Martijn Pallandt (Reviewer 1) for his constructive review. The reviewer's comments are below in black and our responses are in blue. Due to significant changes to the text, not all changes are detailed here, and we direct the reviewer to the full tracked-changes version of the resubmitted manuscript.

*General*

The authors set out to create a unique multispecies dataset of urban fluxes (comprising of $CO_2$, CO, NOx $CH_4$ and N20) to test a new method to partition urban GHG fluxes, specifically into stationary combustion, road transport, and biosphere components. Both this dataset and the methods are novel and of importance since there is a great need for tools to properly partition fluxes in heterogeneous environments. Overall, the writing is clear, the reasoning is sound and the manuscript follows a logical order. Two problems arise during this research. One being unrealistic low values in NOx measurements which are then scaled to local measurements. The other is that the novel partition method struggles in this high flux, high heterogeneity environment, resulting in a mixture of reasonable and unrealistic partition results. The authors are clear on these limitations and suggest linking this method to more components, flux footprints, and temporal explicit emission inventories as potential improvements. Even though the results aren't entirely as envisioned at the start, they are still valuable and serve as an important stepping stone for further research on this or similar partition methods. As an in-situ case study this research sheds a light on seasonal and daily patterns both in observed and ratios of GHG fluxes in Zurich, and is able to characterize this city as a net source of $CO_2$, CO, NOx, $CH_4$ and N20. It was an interesting read and I'm curious how this research further develops.

*Specific*

**20**: Cities leading these efforts if quite a statement, the references here need some work or the claim needs to be modified: the European commission report doi 404ed on my side, the stad Zurich website probably has no place here ->" Informal or so-called "grey" literature may only be referred to if there is no alternative from the formal literature."(and this is just a website not an archived report). A quick scan of the Lwasa executive summary seems to point out that GHG emissions will very likely increase through cities modernization construction, and further urbanization, though they note there is a need for decarbonization for cities.

The European Commission report had an incorrect DOI, this has been updated.

The city of Zurich website reference has been updated to an archived report of the city's most recent net zero 2040 progress report. This is grey literature but appropriate here as it

demonstrates that the City of Zurich has a more ambitious climate plan than exists on the national scale.

The sentence makes two claims: that cities are critical areas for emission reduction efforts, and that many cities have ambitious plans independent from / in addition to those at the national level. We have modified the sentence structure to make it clearer, and softened the latter claim. The new sentence reads:

*Cities are therefore crucial for emission mitigation initiatives (European Commission, 2021; Lwasa et al. 2022), and some cities are already pursuing ambitious plans for reduction of carbon dioxide ($CO_2$) emissions above and beyond national-level plans (e.g., Stadt Zürich, 2024)*

**108**: Is there a particular reason why these months were chosen? A clearer distinction between winter and summer would be observable if June, July and August were chosen instead. Related to this at line 156 /table 2: the winter period is considerably longer at 5 months vs 3 and probably with worse weather, it can be informative to indicate the distribution of (successful) samples between summer and winter.

These months were chosen as a compromise between competing deliverables and milestones of the ICOS PAUL project, namely the desire to have measurements across three pilot cities in sequence (Zurich, Paris, and Munich), to capture cold/warm months, and to fit within the limited project timeline. Therefore the instrument was first installed in Zurich in late July 2022, and needed to move to the following city (Paris) after March 2023 due to project timelines. A longer time series in one city is desirable but was not possible within the 3-year project timeline, measuring 3 cities (including installation of three towers and three campaigns). Table 2 has been updated along with reviewer 2's comments for clarity and to make the seasonal difference in data retention and total preserved 30-minute averaging periods clear.

**165**: It is unfortunate when equipment malfunctions, time dependent in-situ measurements as these cant be redone and you have to work with what you have. However, this section goes fairly quickly from noting that observations didn't match expectations to scaling them up. Some essential steps appear to be missing.

Was the source of the error investigated, (e.g. was it a mechanical or a calibration issue), why did it only affect NOx fluxes, and have similar problems been reported with this instrument? Without identifying the source of the issue, it is hard to justify a correction.

The NOx flux was corrected based on local measurements, but why not perform an actual recalibration/correction with calibration gasses? That would give more certainty in its accuracy than a local measurement which adds several layers of uncertainty such as transport and their measurement errors. It would also allow for additional tests on any drift, to verify that the adjusted slope is stable over time.

I assume that the throughout the paper the adjusted NOx values are used as if they were the actual measurements, and the increased uncertainty was not propagated throughout the rest of the analysis. In either case, please state clearly how this was handled.

The difficulties with the NOx measurements are frustrating and are a significant source of uncertainty that propagates through the study. We agree that these concerns were not sufficiently addressed in the initial submission. We have added additional text to the main body (S. 2.3) which details the quality control and calibration approaches that were used, namely a zeroing tank of $N_2$ was used to zero reactive species NO and $NO_2$ and a co-located flask sampler, air samples from which were analysed by the ICOS Flask and Calibration Laboratory in Jena, Germany, was used to verify good performance for species $CO_2$, CO, $CH_4$, and $N_2O$. Unfortunately the flask sampler is not suited to measuring $NO_x$ and therefore the use of a background reference station was employed as part of quality control.

We suspect two contributions to the low $NO_x$ measurements:

1) losses due to the inlet construction on the order of ~20%. The inlets used in the study are a standard part from LiCOR consisting of rain/dirt cap and approximately 1 m of steel tubing. These inlets are designed for measurements of CO2 and H2O fluxes and it is now clear that they are not desirable for $NO_x$.

We can estimate the losses due to the inlet system based on the subsequent measurement campaign that was conducted in Paris from January - June 2025 following re-calibration and upgrades by the manufacturer. Then testing of the instrument alongside standard $NO_x$ analyzers. The inlet system was identical between both Zurich and Paris campaigns, as was the inlet length (33 m +/- 2 m). The instrument was run co-located with instruments operated by colleagues at the Climate and Environment Sciences Lab (LSCe) in France. Here we found a bias of ~20%, which we attribute to the steel inlet.

2) assumed errors in the spectroscopic setup and retrieval functions accounting for the remaining bias, inclusive of errors associated with the reference measurement station and atmospheric transport to the tower. As noted we are unfortunately unable to independently verify this as the problem was only identified after the end of the field campaign and after the instrument had been returned twice to the manufacturer for various upgrades and repairs.

The large correction applied to the $NO_x$ measurements is not ideal, but we note that it is applied to 8 months of high-quality and standardised measurements collected using a Teledyne T200 chemiluminescent monitor by the Swiss Federal Laboratories for Materials Science and Technology (EMPA) and that we have not simply scaled the fluxes to reach a pre-desired value, but rather corrected the raw concentrations based on a high-quality and regularly-calibrated reference instrument, albeit not in the most perfect circumstances, and re-calculated the fluxes from these data. More information on the reference station measurement program is available in the document below (only in German):

*Technical Report on National Observation Network for Air Pollutants (NABEL). https://www.empa.ch/documents/56101/246436/Technischer_Bericht_2024/0d7b63e5-70a1 -4fba-ad3a-599347447a32*

We have updated the text of this section to incorporate this information, and along with other requests from the reviewer, made additional explicit references to the $NO_x$ uncertainty throughout the manuscript. We hope therefore that while the uncertainties are large, the reader is sufficiently well-informed of these.

Please see the updated section 2.3 for the updated text as well as our response to the second reviewer for individual plots relating to the quality control.

**179**: Why this 4x4 square and not the entire city or a larger part of the tower footprint?

The 4 x 4 $km^2$ was a simplification chosen to cover the majority contribution of the individual 30-minute footprints used in the analysis. Nevertheless, this 4 x 4 $km^2$ box covers 67% of the long-term footprint, representing the majority of the emissions measured at the tower, and also covers the most densely populated portion of the city of Zurich.

Most importantly though, the analysis presented in the paper does not make use of individual footprints: rather the box is used only to extract characteristic sector-specific emission ratios, and further subdivided to determine the variability in those ratios with direction. As the portions of the 70% to 80% ensemble footprint extend largely towards vegetated areas (SW) or similar land uses as within the box (NW, NE), expanding this box does not significantly change the ratios. We have updated Table 3 to include the ratios obtained from considering the whole city, and these fall within the directional variability within the 4 x 4 $km^2$ area.

**Figure 3:** Not essential but a 6th panel in this style with temperatures would be interesting since these would be the main drivers in differences between winter and summer.

Agreed: a 6th panel with air temperature has been added to the figure (now Fig. 4 in the new version).

**301**: If you can conclude based on figure 5 that the ratio can be a mixture of road transport and stationary combustion, why not a combination of respiration and road transport?

We agree that any measurements made can (and likely do) also include a respiration flux of $CO_2$. We are just noting here that in the winter months, the measured fluxes are well-bounded by the inventory references. We have added a sentence to this section to make it clear that a non-fossil $CO_2$ flux from human respiration is present in both seasons.

**350 / figure 6**: This explanation mainly fits the winter period. Summer 2-8 is largely indistinguishable from 6-7 when I look at the figure, and the afternoon rush seems to start at 12 already. Summer seems different from winter here.

Yes this is a fair point: this difference is mostly driven by source area and a greater prevalence of NE winds during the summer during these hours compared to the winter. We have modified this sentence to address winter and summer separately, noting this difference and reiterating the source-area contribution. Nevertheless, this species pair does not show a statistically significant difference between the seasons in aggregate.

**375 /figure 7A**: The nocturnal fluxes would be the yellow area around y0.7 x1.3 just outside the dashed lines? Without a temporal component to this figure, it is not directly clear which the nocturnal NOx fluxes are. Can you clarify.

Yes these fluxes, which form a mode just outside the dashed lines, are what we're referring to. The text has been modified to make this explicit.

**400 and 418**: What can be done about this unrealistic allocation that goes well beyond measured quantities? While the section starting around 427 discusses improvements in general, can you give specific advice on preventing this problem in future cases? Maybe adding more components or limiting allocation to know maxima?

The unrealistic allocation can occur where the measured ratios fall outside the limits defined by the reference ratios from the inventory. One could avoid this entirely by only considering measurements where all ratios fall within the defined reference limits, but this dramatically reduces the number of available measurements (and as we note, treating ratios as constants is probably not sufficient in itself). As we conclude later, the solution is likely improved (measurement-driven) and time-resolved reference ratios.

**453-455**: If I followed the equations in section 2.5 correctly al equations for the final partition components include the NOx term directly or indirectly. Therefore, wouldn't this affect all ratios?

Correct, all partitioning is ultimately affected by the $NO_x$ ratios (or more precisely: the relationship between measured $NO_x$ ratios and the reference ratio definitions). We have modified this sentence to make it explicit that elevated $NO_x$ may lead to over-attribution to road transport of all species.

Did you at some point test the sensitivity of this model to FNOx? Even if you are not certain of your measurements it would give an indication of the impact of such errors.

The sensitivity to $FNO_x$ was tested by scaling the $NO_x$ fluxes and running the model for a series of linear scalings from 0.5 to 1.5 where 1.0 represents the fluxes used through the

manuscript, i.e., those calculated from concentrations scaled against the reference station. The results are presented in the figure below.

[Figure]

As expected the model is sensitive to the NOx inputs as the partitioning of the net flux to source categories is essentially determined by the measured FNOx:FCO along with the reference ratios, and these fluxes are of a similar magnitude.

**456 - 465**: I am somewhat surprised to find new methods and results after the discussion of the model's limitations. And it is not clear to me what you have done here. These two paragraphs and table 6 can benefit greatly from a rewrite to clarify the method and intent. I would advise to add a subsection to the methods section describing this sensitivity analysis in a bit more detail, and discuss these results probably before ~427

Specifically:

1. Table 4 nor its description make mention of a and b. what is exactly being combined here?
2. In the section on the linear mixing model no mention is made of the mmol mol resolution, in which way is it different here?
3. You are testing the sensitivity of the linear mixing model to changes in what exactly? The discussion in the second paragraph also doesn't make clear what sensitivity has been tested here. A typical sensitivity analysis tests a range of values of certain parameters.
4. Continuing with table 6: The first '% of total' refers to table 3 where you have 5 categories and the second 'relative' to only the two listed in this table 6 (sc and rt)? Please clarify.

5. And the model outputs should be compared to the relative column since the model only uses the sc and rt categories? If the model outputs are per wind direction, it doesn't make much sense to compare them to the general reference inventories, wouldn't it be more interesting then to have 5 relative inventory references: all and the 4 directions?
6. In line 496 it is mentioned the sensitivity to reference inputs was tested here?

We agree that this section is more suitable before the assumptions of the model limitations, and we have reorganised/rewritten to try and better clarify what we have done. In response to specific points:

1. The parameters a and b are defined according to Eq. 4-9 and the reference values used from the inventory are those found in Table 4. We have updated Table 4 to make this definition explicit, where before it was implicit.
2. This was poor wording: rather the model was run for every combination of *a* and *b*, within the range found in Table 4, at increments of 0.05 mmol mol-1 (0.1 was from a previous run but provided insufficient granularity for some ratios with smaller ranges, this has been updated). I.e., a set of input ratios was made for a_rt: [3.00, 3.05, 3.10, 3.15 … 4.40], and for a_sc: [0.95, 1.00, 1.05 … 1.40], and the same for b_rt and b_sc. All possible combinations of the inputs were tested and the distribution of model outputs statistically analysed in this section. The wording has been updated to try and make this clearer.
3. Our response to (2) should hopefully make this clear.
4. Correct; we have updated the caption and text to make this more explicit.
5. The relative column is probably more relevant, yes, as it incorporates the assumptions of only 2 combustive source categories. The second part of the question we hope has been addressed by our response to (2).
6. See response to section (2).

Please see the tracked changes version to see all specific changes made to the text.

**468** That would be a valuable continuation of this research, though if you continue with this dataset really aim to get a proper recalibration of the NOx data.

Unfortunately this is not possible, as noted above.

**477-489**: Since not everyone might look at the methods (in detail) a disclaimer on NOx uncertainty in the second or third paragraph is appropriate.

A disclaimer / note on the $NO_x$ correction and associated uncertainty has been added to this section.

**466**: "complex and heterogeneous urban environment may ultimately pose too great a challenge for the application of such a model with fixed emission factors over long periods

of time and large flux footprints." And **501**: " however in the complex and heterogeneous urban environment this information is difficult to exploit on its own."

This is unfortunate, since these are the environments where flux partitioning is especially important. In a homogeneous environment partitioning of fluxes is of lesser importance. Hopefully next steps will improve on this method further.

Indeed, and the following work on this dataset will try to address this problem. The partitioning model developed here essentially asks the question: Can we use measured ratios, reference ratios, and some simplifying assumptions to partition net fluxes even in a complex environment? The results provide a springboard to help us in the next stages to target areas where we can reduce or eliminate the need for certain assumptions as the partitioning becomes more complex to match the real urban environment.

*Technical*

~**427**: not essential but you could put a section break here with everything after ~427 a discussion of the assumptions and where they are met/failed.

After rearrangement of the text in this section we think the progression is more logical and that the discussion flows better without a section break.

**483**: tower..

Fixed.

---

## Author Comment (AC2)

**Response to Reviewer Comment 2 - Erik Velasco**

The authors wish to thank Erik Velasco (reviewer 2) for this thorough and critical review. Below we address each comment individually. Reviewer comments are in black and our responses are in blue. Due to significant changes to the text, not all changes are detailed here, and we direct the reviewer to the full tracked-changes version of the resubmitted manuscript.

Eddy covariance is the best available method for measuring fluxes of non-reactive trace gases, including greenhouse gases and selected pollutant species, between the urban surface and overlying atmosphere. However, its application is complex and limited to urban areas relatively homogenous in terms of land cover, urban morphology, and emissions distribution. In this context, the work presented in this manuscript challenges the application of the EC method by installing an EC flux system on a tall building to measure emissions from the entire city of Zurich, Switzerland.

Fluxes of $CO_2$, $CH_4$, $N_2O$, CO, and $NO_X$ were measured over eight months, three in the summer and five in the winter. The emissions source partitioning was determined using a system of equations based on the flux ratios of $CO_2$, CO, and $NO_X$. For this endeavor, emissions reported in an available bottom-up inventory were retrieved from a square area of 4 x 4 km, with the EC flux system in the center, and were considered correct in order to be used as a reference to solve such a system of equations.

The proposed approach makes sense, but there are a few issues that need to be addressed. The authors must demonstrate that the EC system was able to measure consistently turbulent fluxes, that they were representative of the targeted source area, and that the reported CO and $NO_X$ fluxes were not affected by atmospheric chemistry.

**1**. Installing flux systems on very tall towers (> 100 m) may not be representative of the urban ecosystem. On one hand, tall towers may reach above the top of the collapsing boundary layer at night, causing difficulties in interpreting the flux data collected. On the other hand, if the EC instruments are installed above the upper extent of the inertial sublayer, the development of an internal boundary layer responding to regional-scale land use changes in an upwind region 100–300 times the measurement height may also affect the observed fluxes.

We agree that there can be problems relating to the representativeness and surface coupling of tall-tower measurements. However, with appropriate siting considerations and conservative quality control filtering of measurement periods, the errors and uncertainties associated with tall-tower EC may be reduced. The use of a friction velocity filter, for example, is a common method for minimising periods of surface decoupling.

The quality control used in this work draws on many years of tall-tower eddy-covariance studies and employs best practices according to the literature, relevant citations to which are given in the methods. Unless specific critiques are raised we can only note that while tall-tower EC comes with challenges, we have applied appropriate filtering in accordance with the literature and current best practices of the community.

We believe the aggregated diurnal courses displayed in the manuscript to be well-representative of the city of Zurich based on both footprint extent as well as the good agreement of winter flux ratios with expected ratios from the emissions inventory. This does not remove the possibility that individual measurement periods may contain influence from the concerns raised by the reviewer, but it does minimise their impact on aggregated statistics.

**2.** Representativeness problems may also arise when installing EC flux systems on very tall towers. First, the system would collect fluxes associated with eddies of different sizes and ages, with some originating before the beginning of the averaging flux period and possibly from beyond the city boundaries. Second, as shown by the footprint analysis, the system collected flux data beyond the square area of 4 x 4 km used as a study case.

As we demonstrate below (response to point 5), the estimated average travel time of eddies to the tower is well below the averaging length of 30 minutes used in the flux calculations. With appropriate quality control we have no reason to think that the measurements presented here are fundamentally flawed due to the tower height. The 4 x 4 km area used to obtain reference inventory ratios represents the majority of the flux footprint (discussed below and in response to reviewer 1) and as we emphasise below, ratios do not change significantly outside of this area either.

**3.** Wouldn't it have been more appropriate, both in terms of theory and operation, to have installed the flux system on a much smaller platform if only an area of 4 x 4 km was to be evaluated?

The 4 x 4 $km^2$ box from which inventory reference ratios were obtained was chosen for simplicity, and covers the majority of the built-up core of Zurich and the integral flux footprint (67%).

We note however that we are not only evaluating a 4 x 4 $km^2$ area: we explicitly investigate in this work whether, without the use of individual footprint modelling, we can partition net fluxes from varying source areas and EC footprint sizes. We only use the 4 x 4 $km^2$ area to determine characteristic emissions ratios for the source categories, and also characterise their directional variability for later analysis. To that end it is not necessary that this area extend to the full ensemble footprint extent, but only that it is sufficiently large to characterise the emission ratios. We have calculated the full-city emissions ratios (and added these to Table 4) and in all cases except the $NO_x$/CO road transport ratio (which differs by 0.01 mol $mol^{-1}$) they fall within the previously defined bounds based on wind

direction sector and are therefore already included in the sensitivity testing of Table 6 (Table 7 of resubmission).

While a non-negligible part (33%) of the integral footprint extends beyond the 4 x 4 km$^2$ reference 'box', e.g., to the southwest especially, the box a) already covers the majority of the flux footprint, and b) is large enough to appropriately categorise the emission ratios of each sector. Additional parts of the footprint (such as the SW) are largely vegetation (and therefore irrelevant to stationary combustion or road transport ratios) or (e.g., to the NW) do not add significantly new source sectors or different ratios.

**4.** It is necessary to prove the good performance of the closed-path system. First, the capacity of the MGA7 analyzer to measure the targeted species must be shown. Information on the calibration procedure is required. Second, the spectra and cospectra of the measured variables must be examined to ensure that the flux system was fully capable of measuring turbulent fluxes.

We agree that, as the first eddy covariance paper using this instrument, it is our job to better present the instrument performance.

Regarding measurement accuracy: During the campaign we also operated a flask-based relaxed eddy accumulation (REA) system at the same location. The analyser used fast-switching valves attached to two sampling lines to sample up- and down-draft winds (determined from the same OPEC instrument used in this study) into two separate glass flasks during sampling periods lasting ~1 hour. The concentrations in the flasks were then analysed by the ICOS Flask and Calibration Laboratory in Jena, Germany. More information on the REA system and measurement of concentrations can be found in Kunz et al. 2025 (https://doi.org/10.5194/egusphere-2024-3175). For each REA measurement period the contents of the paired flasks (one updraft flask and one downdraft flask) were analysed and the difference of the flasks ($\Delta\chi = \chi_{up} - \chi_{down}$) was calculated. As the two systems are synchronised in time we can determine the times in which the REA system was sampling the up flask and the down flask, and determine the corresponding average concentrations from the MGA7 as well, and therefore also the $\Delta\chi$ as measured by the MGA7 from the same up- and down-drafts over the REA sampling period. Corrections are applied to account for the response time of the switching valves.

A bias in the MGA7 measurements, which would cause an error in the measured fluxes, can be determined by analysing multiple flask pairs and considering the ratio of $\Delta\chi_{MGA7}/\Delta\chi_{REA}$. If there is no bias in the MGA7 measurements which would affect fluxes, the average of a series of $\Delta\chi_{MGA7}/\Delta\chi_{REA}$ with sufficient signal:noise should equal 1. In total we compare 80 REA periods and show the results below, plotted as a function of the absolute $\Delta\chi_{MGA7}$. This quality control method is available for species $CO_2$, CO, $CH_4$, and $N_2O$. Unfortunately for reactive species NO and $NO_2$ this method is not available. In the below plots one outlier flask was removed from the $CO_2$ and CO analysis where the $\Delta\chi$ ratio was >

3.5 and absolute differences in $\Delta\chi_{MGA7} \sim 0$. Two similar outliers were removed from the $CH_4$ analysis, and no outliers were removed from the $N_2O$ analysis.

As absolute $\Delta\chi_{MGA7}$ approaches 0, the measurement errors of both systems cause significant scatter in the $\Delta\chi_{MGA7}/\Delta\chi_{REA}$ ratios. Therefore, we present both the agreement for all flask pairs as well as for those exceeding an absolute $\Delta\chi_{MGA7}$ above which the relative errors stabilise. Considering all points, the bias in $CO_2$, CO, and $CH_4$ is < 3%, while $N_2O$ shows the largest bias at ~40%. This is explained by the small absolute concentrations, small concentration differences, and large measurement uncertainties. If we consider only samples above the absolute $\Delta\chi_{MGA7}$ threshold, bias is < 1% for all species except $N_2O$, where it is ~5%.

Information on this calibration and QC procedure has been added to the text of the manuscript in section 2.3. While it is also possible to use the individual flasks to calibrate the absolute accuracy of concentrations, this was not performed as the eddy covariance method does not require accurate measurements of absolute concentrations, only that there be no bias in the measurements.

Unfortunately, the flask sampler is not suitable for use with reactive species NO and $NO_2$, and with the lack of an in situ reference instrument, we chose to compare against a high-quality background station under favourable conditions. Please see concerns raised by Reviewer 1 and our associated responses as well to see the changes/qualifiers we have added throughout the paper to ensure the reader is well-informed.

[Figure]

[Figure]

Higher-frequency characterisation of the absolute accuracy of the MGA7 against a high-quality reference instrument was later performed against an EMPA measurement station in Dübendorf, Switzerland, over 2 weeks at 1-minute resolution for all species except $N_2O$. This comparison was only performed 9 months after the Zurich installation and after significant upgrades were made to the instrument, but shows very good agreement (within 2-3%) for all species except NO (7%).

[Figure]

Regarding turbulent spectra: we have added a plot showing the average normalised spectra and co-spectra for each species measured by the MGA7 as well as for the IRGASON and included this as a figure in the manuscript (Figure 1 in the resubmission). Accompanying text has also been added to section 2.2.

Please see the tracked changes version of the manuscript for a complete list of changes made.

**5.** The EC method is limited to non-reactive species or species with low chemical reactivity, particularly in highly reactive atmospheres like those in urban areas. Measured fluxes of $NO_x$ and CO, especially the former, may not represent emissions at surface level, since chemical reactions might drastically lower their abundance before reaching the height at which the EC instruments were installed. Consider the lifetime of these species by calculating their reactivity with the hydroxyl (OH) radical and the time it takes an air parcel to travel from the surface to the top of the tower. If OH data for Zurich are not available, use data reported for other cities to get a rough estimate (e.g., Price et al., 2025; https://doi.org/10.1038/s43247-025-02308-y) ☐. The time it takes an air parcel to reach the top of the tower can be determined using the standard deviation of the vertical wind speed fluctuations.

We do not believe that atmospheric reactivity between the surface and the measurement height is a significant concern in this study. As suggested by the reviewer, the travel time from surface to measurement height was calculated using a simple statistical approximation with a tower measurement height **z** of 112 m agl and the standard deviation of vertical wind speed fluctuations, i.e., travel time $t = \sigma w'/z$. The results, after applying all filtering described in the paper, are shown in the following figure in minutes:

[Figure]

The median travel time is ~3.5 minutes and the maximum is ~19 minutes. Less than 2% of included measurement periods have a travel time greater than 10 minutes.

In contrast, the lifetime of atmospheric $NO_x$ in the urban boundary layer is on the scale of many hours, rather than minutes. OH data are not readily available for Zurich, however the authors were not able to find any evidence in the literature supporting an urban $NO_x$ lifetime of < 10 minutes, with typical reported values being 10 - 50 hours. Even if we assume a very short lifetime of atmospheric $NO_x$ from the literature, it would only imply a 1-2% loss at measurement height, which is a comparatively minor error for the EC method.

A search of the literature indicates that this subject has been evaluated elsewhere and there are multiple examples of EC $NO_x$ fluxes in the literature, both on (tall) towers as well as aircraft:

- Vaughan et al. (2015) https://doi.org/10.1039/c5fd00170f (aircraft; 360 m agl; published in Faraday Discussions, Royal Society of Chemistry)
- Lee et al. (2014) https://doi.org/10.1021/es5049072 (tower; 190 m agl; Environmental Science and Technology)
- Vaughan et al. (2021) https://doi.org/10.5194/acp-21-15283-2021 (aircraft; 330-360 m agl; Atmospheric Chemistry and Physics)

Many more references on shorter (< 100 m) towers are also available. Based on this collective evidence, we reject the reviewer's claim that $NO_x$ and CO fluxes can not be measured via the EC method at all due to atmospheric reactivity. The manuscript has been updated to identify the estimated travel time more specifically based on the above plot and relevant references have been added.

**6.** Eddy covariance flux towers are used to evaluate the accuracy of emissions estimated using bottom-up approaches, not the other way around. If the emissions reported in the inventory are presumed to be correct, what is the study's purpose? This reviewer has used eddy covariance flux measurements to evaluate the accuracy of gridded emission inventories of $CO_2$ (Velasco et al., 2014; https://doi.org/10.1016/j.atmosenv.2014.08.018 ) and selected volatile organic compounds (Velasco et al., 2005, https://doi.org/10.1029/2005GL023356; Velasco et al., 2009, https://doi.org/10.5194/acp-9-7325-2009) . These studies could provide insight on the use of flux towers for such a purpose.

Evaluation of the accuracy of bottom-up emission inventories are *one* use of EC towers but far from the *only* one. Evaluation of GHG fluxes using the EC method is the topic of a number of papers currently submitted or in preparation as part of the ICOS cities project. The purpose of the current study, however, is to a) present urban fluxes of not only $CO_2$ but also co-emitted species CO and $NO_x$ as well as greenhouse gases $CH_4$ and $N_2O$ (section 3.1); b) to investigate diurnal and seasonal patterns in the ratios of fluxes of $CO_2$, CO, and $NO_x$ (section 3.2); and c) to evaluate whether co-emitted species can be used to partition net fluxes to emission source categories under certain assumptions (section 3.3). We believe this is clearly stated in the title, abstract, and introduction.

Critically, c) is only one part of the study and is only pursued after demonstrating reasonable agreement with the emissions inventory in b). Further, the assumption of accuracy on the part of the inventory is limited only to the ratios of specific species, and we do not make assumptions that the absolute magnitudes are correct, nor that temporal profiles or spatial distributions are accurate, all of which are separate questions.

The objectives of the study are made clear in the title, abstract, and introduction, and in all cases the authors feel it is made clear that the scope of the study is not an inventory evaluation. The addition of specific research goals added at the request of the reviewer also helps to make this clearer.

Specific comments

**P1, L2.** Define all abbreviations and acronyms in the abstract and main text when they appear for the first time.

Fixed to define $CO_2$.

**P1, L5.** Does '1.47x' mean 1.47 times the variable 'x'? Write '1.5 times larger …' Review the entire text and correct this spelling error.

Changed as requested.

**P1, L5-8**. The main text indicates that the differences observed between summer and winter were statistically significant; please also indicate this here. Include a variability metric.

Unfortunately, abstracts for this journal have a hard limit of 250 words and to incorporate the reviewer's previous requests we have already had to remove other important information from the abstract. It is simply not possible to add a variability metric for each species and season combination (addition of 30+ words) as well as explicitly state which differences were statistically significant or not, and still retain other necessary information about the study in this limited space.

If readers are interested in the variability they may read the rest of the paper, and the authors believe that the statement that differences for $N_2O$ were *not* statistically significantly different should be sufficient implication for the reader that the other differences are.

**P1, L19**. … 70% of global energy-related GHG emissions … directly and/or indirectly?

Per the reference: *this estimate is based on consumption-based accounting, including both direct emissions from within urban areas, and indirect emissions from outside urban areas related to the production of electricity, goods, and services consumed in cities.*

**P2, L22** Define bottom-up approach.

A brief definition of a bottom-up approach has been added.

**P2, L23**. What do you mean by temporally coarse (yearly, monthly, daily, hourly)? Consider that many cities estimate gridded emissions for air quality forecasting on an hourly basis.

This varies based on the method employed, however those inventories which produce hourly estimates of emissions tend to do so by partitioning yearly emissions based on activity/proxy data, with large associated uncertainties in the temporal allocation schemes, as already mentioned in the manuscript (line 24 of original submission). We already make clear in the manuscript that we are working with the yearly emissions inventory (from which hourly estimates may be forecast, but only based on simple scaling factors which do not adequately account for real-world variability).

**P2, L28**. To avoid possible useless discussions, replace 'the only method for' with 'the best available method'.

Changed as requested.

**P2, L47**. Provide more examples, as this reference only reports EC flux measurements in one city. Alternatively, refer to review papers on the EC application over urban surfaces: Feigenwinter et al., 2012 (https://doi.org/10.1007/978-94-007-2351-1_16 ☐) or Velasco et al., 2010 (https://doi.org/10.1111/j.1749-8198.2010.00384.x) ☐.

The study referenced in the manuscript presents EC fluxes from one city, but also provides a thorough overview of all urban and suburban EC studies of $CO_2$ fluxes from the last ~25 years (see Table 1 in the reference). This not only provides the same references of Feigenwinter et al. (2012), but also provides many additional citations for the reader in the 10+ years since that publication.

 We have added the suggested overview Feigenwinter 2012 reference to this sentence.

**P2, L47**. The EC flux method has also been used to measure fluxes of selected volatile organic compounds and aerosols to evaluate the accuracy of bottom-up emission inventories.

Agreed, and we note in this sentence that the method can be applied to any scalar. It is still true however that urban EC has overwhelmingly focused on $CO_2$. The rest of this section only considers works that have measured gases which will be discussed in the current study, and is not intended to be a comprehensive review of urban EC or all scalars that have been measured via EC. VOCs and aerosols are outside the scope of the study and not relevant for discussion here.

**P2, L50.** Carbon monoxide is a criteria pollutant for regulatory purposes, and therefore environmental agencies must report its ambient concentrations on an hourly basis.

Agreed, and we noted in this sentence that its role as a common urban pollutant requires its measurement and reporting on a 30-60 minute basis. We have tried to improve the wording for clarity.

**P3, L67-74**. You are listing what was done in this study, but you are missing the reason. State clearly the hypothesis to be tested and the questions to be answered.

We have reformatted this paragraph to re-state it as a series of research goals and hereby added the reason.

**P3, L75**. A more detailed description of the monitored footprint is required, in this case, the entire city of Zurich. Provide information on building characteristics and energy consumption, indicate the types of industries and vehicle fleet characteristics, include population density, describe the city's vegetation, and provide tree density. Also, describe the climate throughout the year.

We have expanded this section to include the requested information, where available, using data from the city of Zurich. Air temperature as observed is added later as a panel in Fig. 4.

**P3, L83-84**. Was 17 m above roof level sufficiently high to prevent wind flows from obstructing turbulent flux measurements? Demonstrate that the building in which the system was mounted did not create disturbing flows.

It is not possible to demonstrate (nor realistic to assume) that the building and tower had no influence on the measured flows: any measurement tower will influence the measurements to some degree. We filter for obstructing wind directions and remove very high angles of attack (this note was missing in the original submission and has been added) to try and minimise the distorting influence of the building.

The figure below shows statistics describing the wind distortion using all IRGASON data with wind speed above 1 m/s for a period of two years. The data are binned per wind direction (20 degree bins). The above plot shows the boxplots of the normalized turbulent kinetic energy and the plot below the mean angle of attack.

[Figure]

In the upper panel above plot we can see the expected wind distortion from the mast in the 90° bin and possibly two other slight peaks in norm. TKE from S and N which could be attributed to the building structure and/or the nearby high buildings. However, the values of the normalized TKE are sufficiently low and do not indicate considerable distortion. Highly distorted wind sectors would show much higher values and increased peaks compared to non-distorted sectors. A good example is given in Fig. 3 of Järvi et al. 2018 (https://doi.org/10.5194/amt-11-5421-2018), where normalized TKE peaked at average values near 2.

The lower panel of the plot shows that there is a consistent pattern in the mean angle of attack, with near horizontal wind from S and slightly inclined upwards (10 – 15 degrees) from N. This pattern is most probably the effect of the building itself on the mean wind flow streamlines since the mast is located at the N side of the building. The wind flow is expected to diverge upwards over the roof and we can observe that when the wind is coming from N since the instrument is closer to the windward side of the building. When the wind is coming from S, this flow displacement effect has already nearly disappeared since the instrument is at the leeward side. Overall, the measured angle of attack values are sufficiently close to horizontal indicating that the instrument is not experiencing significant flow distortion that might affect the eddy covariance observations. A good example of angle of attack distortion due to bad instrument placement on a high-rise building is shown in the respective section (Flow Distortion, Fig. 16.5) of Feigenwinter et al. 2012 (https://doi.org/10.1007/978-94-007-2351-1_16).

We refer the reviewer further to the similar discussion in Lan, et al. 2024 (https://doi.org/10.5194/amt-17-2649-2024), especially Fig. 4-b, which presents results from this same tower and discusses possible building influence.

**P3, L84-87**. But the footprint analysis indicates that the EC system measured fluxes from a much larger radius. In the best scenario, a 1.5 km radius will cover 30-40% of the observed source area. This reviewer does not believe that the characteristics of the district where the EC system was installed are representative of the overall footprint that was observed.

The mean building height within the 4 x 4 km$^2$ box is nearly identical: average of 14 m +/- 8.5. This covers the majority of the built-up area of the city and the majority of the flux footprint. The mean building heights of Zurich do not change significantly at the farthest extents of the city in a way that would fundamentally call into question the validity of these building height descriptions nor the results obtained. The building heights described in the original manuscript were used as surface parameterisations in EddyPro and therefore it is reasonable to the authors to leave that building height description in the manuscript.

**P4, L88**. Include the instruments' accuracies and uncertainties as provided by manufacturers in Table 1.

Where provided by manufacturers, these have been added to Table 1. Note however that manufacturers do not provide precisions or accuracies for all variables.

**P4, L88**. How do the $CO_2$ concentrations and fluxes measured by the open-path and closed-path flux systems compare?

The fluxes calculated using both systems agree well:

[Figure]

The raw concentrations between the instruments were not compared in detail as the accuracy of the absolute measurement is not particularly important for EC calculations, so long as there is limited bias. The concentrations as later compared to a reference instrument are shown above in our response to (4).

**P4, L90-100**. Were both instruments periodically calibrated? Failure to do so may result in a serious omission in the study.

The IRGASON was calibrated before the campaign on 6 July 2022 with two $CO_2$ span gas cylinders and one $N_2$ zeroing cylinder. The instrument was not re-calibrated for the duration of the campaign, however given the continued good agreement between IRGASON and MIRO we do not expect significant drift to have occurred during the 8 months.

The MIRO instrument was installed immediately after delivery from the manufacturer. During the campaign the reactive species NO and $NO_2$ were regularly zeroed against a reference tank and the other species were compared against high-quality measurements obtained from a modified ICOS flask sampler and analysed at the ICOS Flask and Calibration Laboratory in Jena, Germany. Please see our responses above for more information on the accuracy of the CPEC analyser as well as our response to similar concerns raised by reviewer 1. Relevant information has been added to the manuscript, please see the tracked changes version for a full list of changes.

**P4, L103**. FFP flux footprint?

"FFP" here is "flux footprint parameterisation" - we have simplified it to just "flux footprint model".

**P4, L103**. Which source did you use to get the boundary layer height that the footprint model requests as input data?

The boundary layer height was taken from ERA5 reanalysis data, and a conservative threshold was applied to remove periods in which the BL height was too low. More information on the footprint modelling is available from ICOS Cities Deliverable 2.6 available from the project website (https://www.icos-cp.eu/projects/icos-cities/deliverables).

**P5, Fig. 1**. The 30% footprint contour looks suspicious. There are two lines in the west sector, but none in the east sector.

The footprint is compiled based on QC-passing periods and therefore the large excluded wind direction band can create some unintuitive artefacts around those wind directions. We have updated the figure caption to make it clear that these wind directions were excluded, but the footprint itself is reasonable given a multi-modal wind regime and significant quality control filtering.

**P6, L110-112**. Are there regulations in place about domestic heating according to the time of year?

To the knowledge of the authors there are no regulations in place regarding domestic heating.

**P7, L126** Open-path instruments and sonic anemometers do not work during rain events. Thus, periods affected by rain must be excluded for further analysis.

The sonic anemometer used here has water wicks on the transducers and while rain events *can* cause the instrument not to function it is not necessarily true that the instrument fails to function during all rain events. The open-path gas analyser data was filtered based on signal strength (though the open path analyser is not otherwise used in this work) and the sonic anemometer diagnostic codes were used to filter periods in which the sonic anemometer did not work due to water droplets or water accumulation.

More information on the behaviour of the IRGASON in wet conditions can be found in the Campbell Scientific User Manual: https://s.campbellsci.com/documents/ca/manuals/irgason_man.pdf

**P7, 131-134**. How do you define a spike? See Schmid et al. 2000 (https://doi.org/10.1016/S0168-1923(00)00140-4) ☐.

We define a spike according to Mauder et al. (2013) while allowing multiple consecutive (>3) outliers. This is a fairly conservative approach and does not contradict Schmid et al. (2000). Urban EC is complicated by possible point sources and very inhomogeneously distributed sources when compared to work over homogenous surface covers like, e.g., that of Schmid et al. (2000).

The validity of common despiking routines is outside the scope of this paper but a topic discussed in S. 2 of Lan, et al. 2024 (https://doi.org/10.5194/amt-17-2649-2024) for this and other ICOS Cities towers.

**P7, L135-137**. Did the sampling tube and response time of the closed-path analyzer dampen the turbulent signal? Did you examine the spectra and cospectra of the measured variables to determine the influence of potential attenuations. You need to do it to demonstrate that the flux system is capable of measuring turbulence fluxes of trace gases via the EC method.

Yes: the response time of the instrument as well as the tube geometry affected the turbulent signal, and (co)spectra were examined to determine that the instrument was able to measure turbulent signals. We agree with the reviewer that this information should have been provided in the original submission as this manuscript presents the first EC measurements from this instrument in the literature.

We have therefore added an additional figure (Figure 1 in the resubmission) and text describing the performance of the instrument as well as spectral attenuation caused by the tube/setup. In each case the (co)spectra from the open-path IRGASON instrument is also shown for comparison, from which attenuation in the tube can be determined. Please see the tracked changes version of the resubmission to see specifically what was added.

**P7, L140**. Explain what each flag indicates.

We have updated the text to state

*Initial quality control was performed using the 0-1-2 flag system of Mauder and Foken (2004) in which data are categorised as 0 - best quality data suitable for fundamental research, 1 - acceptable quality data for observation programs, and 2 - low quality data.*

A detailed breakdown of what percentages of variance and combination of individual test flags is associated with the integral flag of Mauder and Foken (2004) is not given here as these are well-established techniques in EC QC and further details can be found in the sources provided.

**P7, L146-147**. This is unfortunate. According to Figure 1, this sector covers the most urbanized area of Zurich

Fortunately, as evidenced by the gaussian lateral component of the footprints, this sector is not completely lost from the measurements despite filtering mean wind directions from such a large sector. We have updated the figure caption to explicitly note that this is post-QC filtering. While some of the densely populated portion of Zurich east of the river is not included, the SE wind sector contains a substantial coverage of the dense city core.

**P7, L154-155**. How large was the storage flux?

A note on the storage flux (average and IQR) for each species was added to this sentence.

**P8, Table 2**. Instead of listing the number of averaging periods, show the percentage of averaging periods used for further analysis for each climatological season analyzed. You may add the percentage of periods removed due to instrument problems and calibration, rain events, or failure to reach the flag 0 and u* thresholds. Note, periods with low u* usually do not meet the stationarity test.

We have updated Table 2 to focus on the percentage of retained periods and separated the results by season.

**P8, L156-158**. The study covered three months in late summer and early autumn and five months during the winter. Furthermore, 40-50% of the averaging periods were excluded for further analysis due to quality issues. Was the amount of data collected sufficient to draw conclusive results? Could the difference between data collected in summer and winter be a determining factor in drawing conclusions?

The data retention during this campaign of 50-65% is average to above average for urban eddy-covariance studies as discussed in the manuscript, and this percentage is fairly consistent between the two aggregated seasons as shown in the updated Table 2. We agree the representativeness of the measurements would be improved if we had a full year's measurements, however the project timing did not allow for this (see response to reviewer 1). For this integral study we therefore aggregate measurements to average diurnal courses.

**P8, L167-170**. Check the method used by the $NO_2$ monitor used as a reference. Measurements of $NO_2$ from standard chemiluminescence monitors equipped with molybdenum oxide converters may suffer from interferences and end up reporting concentrations exceeding up to 50%. See Dunlea et al. 2007 (https://doi.org/10.5194/acp-7-2691-2007) ☐ .

The reference station uses a Teledyne T200, a chemiluminescent monitor. Unfortunately we are not able to quantify or correct possible interferences in these measurements. The reference measurements are carried out by the Swiss Federal Laboratories for Materials Science and Technology (EMPA) as part of a national measurement network and with regular, robust calibration and quality control. These measurements are considered to be of

very high quality. If the reviewer is interested, the calibration and QC protocols are extensively documented in the monitoring network's technical report linked below (only available in German).

*Technical Report on National Observation Network for Air Pollutants (NABEL). https://www.empa.ch/documents/56101/246436/Technischer_Bericht_2024/0d7b63e5-70a1 -4fba-ad3a-599347447a32*

**P8, L175**. What is the temporal resolution of the emissions inventory (hours, days, months, years)?

The temporal resolution of the inventory was 1 year (2022). This is stated in the sentence:

*An emission inventory for 2022 was created by the city of Zurich on a GIS format which provides yearly total emissions of various species and pollutants for the city of Zurich (Brunner et al., 2025).*

**P9, Table 3**. Use Mg as a unit. Tonnes are not accepted by the International System of Units.

All instances of tonnes have been updated to SI units as requested.

**P9, L179**. I would say the 4 x 4 km grid cell designated to investigate flux partitioning covers no more than 60% of the observed source area. Now, if the study was limited to that area of the city, was it necessary to install such a tall tower? A much smaller flux tower would have been more effective and would have covered the target area much better.

The 4 x 4 km$^2$ cell is a simplification, albeit one that covers a sufficient portion of the built-up area of the city, and the majority (67 %) of the integral campaign footprint. As noted elsewhere in this response, the full-city ratios are already well-described by this box and it is therefore a reasonable simplification: As we note above, this area is only used to characterise emission ratios from individual source categories; it does not need to perfectly match the extent of the footprint. Future studies from the ICOS Cities project that use a per-footprint inventory comparison will necessarily make use of the full extent.

**P9, L181**. Perhaps 'anthropogenic $CO_2$ emissions' instead of 'non-respiration $CO_2$ emissions'.

This becomes a question of definitions. In this sentence we are separating anthropogenic emissions into those caused by combustion processes and those caused by respiration. Rewriting as "anthropogenic emissions" does not make this distinction as it would include human respiration (in our opinion). We have not changed this wording.

**P10, L200**. Do these ratios imply that the emissions estimates in the inventory are essentially correct?

It is one of the assumptions of the model that the per-sector emission ratios are well characterised. Note that this doesn't necessarily require the absolute emissions estimates to be correct, nor does it require that all sectors in the inventory be well-characterised. Therefore while our measurements suggest that the stationary combustion and road transport ratios are well characterised, it does not imply that the emissions estimates of the inventory are correct per se. These subjects are already discussed in Section 3.3.

**P10, L205-210**. You need to account for NOx lifetime and reactivity throughout the diurnal course.

Please see our response to point number 5 above.

**P12, Fig. 3**. Were there differences between weekdays and weekends? The analysis must account for emission variations based on the day of the week. Studies in other cities have found substantial differences.

There are significant differences between weekdays and weekends for the three primary species analysed in this work. We have updated Fig. 3 (now Fig. 4 on resubmission) and updated the text in section 3 to incorporate this.

**P12, Section 3**. This section is lengthy and tough to read. Many of the results might be better presented in a table or figure. Avoid filling the text with numbers.

We have summarised these data in a new table (Table 5 in the resubmitted manuscript).

**P12, Section 3.1**. Add a variability metric to all figures.

The IQR has been added as a variability metric to the new table.

**P13, L247-249**. Why?

We have added a possible explanation to the sentence.

**P13, L260-261.** Why?

This is a good question, and one which we do not know the answer to. The differences are statistically significant, but only weakly so ($0.01 < p < 0.05$).

**P14, Table 5.** Add a variability metric. You may additionally provide observed fluxes based on wind sector.

We have added the IQR in brackets to the table.

**P14, Fig. 4**. Nice figure!

Thank you.

**P15, L290-291**. This is not true for the case of reactive species such as NOx and CO.

Please see our previous answer regarding reactive species.

**P15, L299-302**. This reviewer does not find such a strong agreement when considering Fig. 5 as a reference. He observes an important variability in the measured ratios and a significant difference against those derived from the emissions inventory. To arrive at this statement, it is necessary to provide the hourly ratios obtained from the emissions inventory.

We find this statement confusing. There is indeed variability within the measurements, but the bounds of the average hourly flux ratios (which we are discussing) are well defined by the inventory reference ratios. That there is clear variability *within* those bounds is also not disputed. We have also stated already in the paper that we are working with ratios determined from a yearly inventory, and we do not have footprint-scale hourly ratios in this study.

This sentence says nothing more than the upper hourly average ratio of $NO_x/CO_2$ (as measured) is well characterised by the inventory ratio for road transport, and that the lower hourly average ratio (as measured) of $NO_x/CO_2$ is well characterised by the inventory ratio for stationary combustion. We have modified the sentence to make it explicit that we are discussing the median hourly measured ratios.

**P16, L327-328**. Similar to the previous comment. Comparing hourly ratios from field observations with daily ratios from the emissions inventory does not lead to this conclusion.

Similar to the previous comment, we feel that the reviewer is misunderstanding what our conclusion is. We do not have daily ratios for comparison, but rather yearly. The plots make it clear that the upper and lower average hourly ratios are well-bounded by the emission inventory source categories of stationary combustion and road transport. We have modified the sentence to make it explicit that we are discussing the median hourly measured ratios.

**P17, L355.** Perhaps 'seasonal changes in FCO2'.

Agreed, we have updated as suggested.

**P18, Fig 7b.** A panel may be presented for each case. It is a bit confusing to identify the histogram corresponding to each ratio and season.

We have updated the figure as requested.

**P19, L387-389.** The type and characteristics of the vegetation determine this. Is Zurich's vegetation similar to that in Indianapolis, USA?

We have modified the sentence to be more reserved: we are simply pointing out that we do not rule out the possibility of non-zero photosynthetic activity during Nov-Mar during individual 30-minute averaging periods.

**P19, L394-396.** Are these differences significant compared to those reported in the emissions inventory? If there is a difference, might it be due to differences in the accounted footprint or modeled area?

These differences are already discussed along with possible explanations in the following sections (see, e.g., the paragraph starting at L420 in the original submission and following discussion).

**P20, Fig. 8b.** For $NO_x$, the stationary combustion sector's contribution looks suspiciously low compared to the total flux of CO and $CO_2$.

We agree that this is "suspiciously low" and already note that the partitioning of the model is "sensitive to reference inventory ratios and can produce improbable and impossible results". We explicitly note in the original version of the manuscript that the model partitioning significantly underestimates the contribution from stationary combustion, and that "the mismatch in the observed stationary combustion ratios shown in Fig. 8-A is a likely source of poor partitioning of $FNO_x$ into stationary combustion.

**P20, Fig. 8c.** The biogenic $CO_2$ uptake during winter looks suspiciously intense. Is Zurich's vegetation evergreen?

We agree that this is "suspiciously intense" and already note in the following sentences of the original manuscript that it is unrealistic. In the original version of the manuscript we explicitly state that the modeled $FCO_{2,bio}$ in winter is unrealistic, and discuss possible explanations.

**P21, L421-426.** Could this be related to differences between the flux tower's footprint and the 4 x 4 km area extracted from the emissions inventory?

No: we do not expect that a mismatch in the footprint and inventory-derived ratios is responsible for this partitioning. As we note elsewhere in this response, the ratios do not dramatically change when incorporating the full city. Instead we expect the mismatch to be due to violations of the assumptions elaborated on in this and the following paragraphs.

**P21, L431.** This assumption was clearly not met using such a tall EC flux tower.

Indeed, and it is arguable whether this assumption is ever met in urban areas. However, we already address these concerns in the following sentences in the original manuscript.

**P21, L432.** Please provide examples of 'biospheric' emission sources of CO and $NO_x$ within Zurich's built-up area.

We do not expect significant biospheric emission sources of CO and NO$_x$ within Zurich. This is already discussed in the following sentences of the original manuscript, though we note that assumption 4, referenced here, explicitly refers to the "total flux" and not only emissions, as the reviewer has asked.

**P22, Section 4.** Conclusion should not be a results summary. You may add a section summarizing main findings.

This section contains both a summary as well as what the reviewer considers appropriate conclusions. We note that this formulation is common and acceptable to the editors of Atmospheric Chemistry and Physics, e.g., from the most recent publications on the website, the conclusions section begins with a brief summary of the article and the results:

https://acp.copernicus.org/articles/25/5773/2025/

https://acp.copernicus.org/articles/25/5977/2025/

https://acp.copernicus.org/articles/25/5959/2025/

We have modified the heading to be "Summary and Conclusions" for clarity, but have not modified the content.

**P23, Table 6**. Are these results for the entire 8-month measurement period, or just the summer or winter periods?

These results are for the entire measurement period and this information has been added explicitly to the caption.

**P23, Table 6.** Explain the potential reasons for the negative CO and NOx fluxes reported in this table.

These occur for the same reason already discussed in the original manuscript re: Fig. 8 (original manuscript) or Fig. 9 (resubmitted manuscript), namely that measured flux ratios exceed the ratio limits predicted by the inventory. We have added a sentence to make this explicit.

**P24, L499-506.** This is the type of statement that should be in the conclusion section.

This is the type of statement that **is** in the conclusion section. We feel that the concerns about the conclusions have been addressed in the response to the comment about "P22, Section 4", and that we have written a conclusion that is acceptable to the editors of ACP..